# Global, regional, and cryptic population structure in a high gene-flow transatlantic fish

Eeva Jansson[1], Ellika Faust[2]*, Dorte Bekkevold[3], María Quintela[1], Caroline Durif[4], Kim Tallaksen Halvorsen[5], Geir Dahle[1], Christophe Pampoulie[6], James Kennedy[6], Benjamin Whittaker[7], Laila Unneland[1], Søren Post[8], Carl André[2‡], Kevin A. Glover[1‡]

1 Institute of Marine Research, Nordnes, Bergen, Norway, 2 Department of Marine Sciences - Tjärnö Marine Laboratory, University of Gothenburg, Strömstad, Sweden, 3 DTU-Aqua National Institute of Aquatic Resources, Technical University of Denmark, Silkeborg, Denmark, 4 Institute of Marine Research, Austevoll Research Station, Storebø, Norway, 5 Institute of Marine Research, Flødevigen, His, Norway, 6 Marine and Freshwater Research Institute, Hafnarfjörður, Iceland, 7 Department of Biosciences, Centre for Sustainable Aquatic Research, Swansea University, Swansea, United Kingdom, 8 Greenland Institute of Natural Resources, Nuuk, Greenland

☯ These authors contributed equally to this work.
‡ CA and KAG also contributed equally to this work and share last authorship.
* ellika.faust@gmail.com

**Data Availability Statement:** Raw sequences from 2bRAD sequencing are available on NCBIs Sequence Read Archive in the BioProject PRJNA858951. SNP data can be found in the

## Abstract

Lumpfish (*Cyclopterus lumpus*) is a transatlantic marine fish displaying large population sizes and a high potential for dispersal and gene-flow. These features are expected to result in weak population structure. Here, we investigated population genetic structure of lumpfish throughout its natural distribution in the North Atlantic using two approaches: I) 4,393 genome wide SNPs and 95 individuals from 10 locations, and II) 139 discriminatory SNPs and 1,669 individuals from 40 locations. Both approaches identified extensive population genetic structuring with a major split between the East and West Atlantic and a distinct Baltic Sea population, as well as further differentiation of lumpfish from the English Channel, Iceland, and Greenland. The discriminatory loci displayed ~2–5 times higher divergence than the genome wide approach, revealing further evidence of local population substructures. Lumpfish from Isfjorden in Svalbard were highly distinct but resembled most fish from Greenland. The Kattegat area in the Baltic transition zone, formed a previously undescribed distinct genetic group. Also, further subdivision was detected within North America, Iceland, West Greenland, Barents Sea, and Norway. Although lumpfish have considerable potential for dispersal and gene-flow, the observed high levels of population structuring throughout the Atlantic suggests that this species may have a natal homing behavior and local populations with adaptive differences. This fine-scale population structure calls for consideration when defining management units for exploitation of lumpfish stocks and in decisions related to sourcing and moving lumpfish for cleaner fish use in salmonid aquaculture.

## 1 Introduction

Understanding population demographics and population genetic structure is important for effective management and sustainable exploitation of wild species. Globally, it has been

supplementary material S3 File for the genome-wide data set and S4 File for the targeted data set. Metadata is provided in S2 File. The original protocol and scripts can be found at https://github.com/z0on/2bRAD_denovo modified scripts and pipeline can be found at https://github.com/ellikafaust/2bRAD-cleaner-fish.

**Funding:** This study was funded by the Norwegian Ministry for Trade, Industry and Fisheries, the Swedish research council FORMAS and the European Regional Development Fund (Interreg project "Margen II") The funders had no role in study design, data collection and analysis, decision to publish, or preparation of the manuscript.

**Competing interests:** The authors have declared that no competing interests exist.

estimated that 34% of the assessed fish stocks are currently overfished, and another 60% are fully utilized [1]. Stock assessments have traditionally been based on changes in abundance, catch and demography (e.g., size, age, mortality and reproduction), but to be accurate, they need to account also for population genetic structure and connectivity patterns among populations, and their underlying drivers [2–4]. Therefore, and where available, knowledge of population genetic structure should be included in management regimes [5–7]. This applies also to marine species that usually display wide distribution ranges and large census population sizes, factors that often are considered to lead to weak population structure [8]. Genetic studies on marine fish have shown that, I) the effective population size reflecting population resilience may be only a fraction of the census size [9, 10], II) the rate of effective migration might be lower than anticipated also in species with pelagic eggs and larvae [11], III) populations often encompass local adaptations [12–15], and that, IV) existing management units often misalign with population genetic structure and barriers to gene-flow [6, 16, 17]. Moreover, as populations are temporally and spatially dynamic entities, it is important that the status of species and their populations is monitored and regularly updated, especially when emerging methods and approaches can provide new insights [18–22].

Lumpfish, *Cyclopterus lumpus*, is a cold-water marine species [23] distributed throughout the North Atlantic [24] (Fig 1) that has a long history of commercial exploitation. Fishing occurs almost exclusively when lumpfish migrate to coastal areas to spawn. At first the fishery was only small-scale and targeted males to be salted, smoked or dried for human consumption

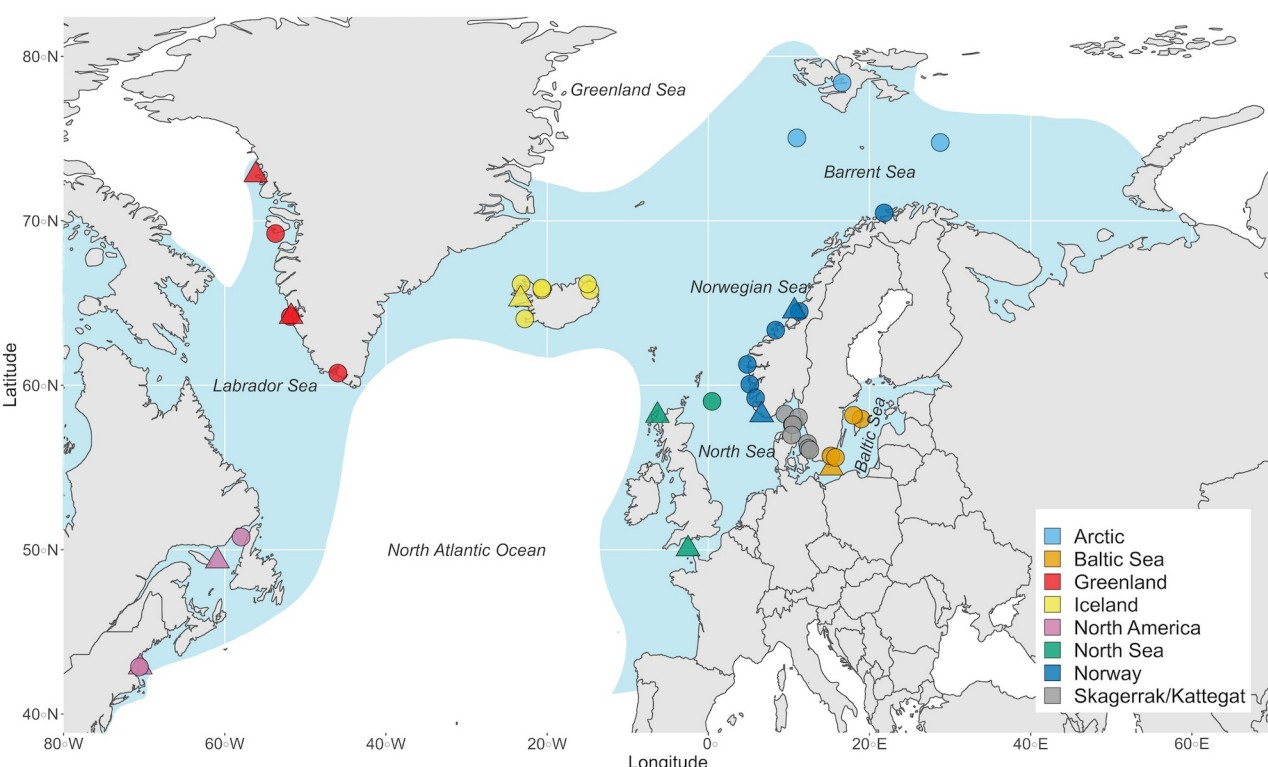

**Fig 1. Distribution map of sampling localities of lumpfish (*Cyclopterus lumpus*) across the North Atlantic Ocean.** Blue background indicates the approximate distribution area for lumpfish. Triangles indicate samples with sequencing data that were used to explore genome-wide population genetic patterns and to develop SNP markers. Circles indicate samples with SNP genotype data only. Colors indicate geographic regions as described in Table 1. Fish from three mid-sea samples, the Barents Sea, Svalbard, and the North Sea, were collected on research cruises covering a wide area. For these, the point shown on the map is the midpoint of all collection sites (for details, see Figure 2A-2D in S1 File).

[24, 25]. Later on, and towards the end of the 1940s, commercial scale exploitation of lumpfish females for their roe was initiated, and is now amounting in many thousand tons per year [25]. Recently, smaller scale fisheries have been established to capture mature individuals to serve as broodstock for juvenile lumpfish production that are in turn used as cleaner fish to remove parasitic sea lice in salmonid aqua/mariculture [26–28]. The use of farmed lumpfish has been expanding rapidly in recent years, and in Norway alone, over 40 million lumpfish are sold annually as cleaner fish to salmonid farms (Figure 1 in S1 File). This use of lumpfish in aqua-culture raises several concerns. First, while the industry is gradually moving towards closed production cycles, the majority of broodfish are still of wild origin [28], adding further pres-sure on wild populations with often unknown status. Second, escapes of translocated lumpfish from net pens in salmonid farms represent a potential threat to local lumpfish populations, such as introduction of non-local genetic variants and/or diseases [29]. Escape of Atlantic salmon (*Salmo salar*) is very common [30, 31], and escape of other species of cleaner fish from commercial salmon farms has already been documented [32–34].

Adult lumpfish are solitary and inhabit the upper 50m of the water column [24] in offshore waters. Lumpfish are generally found in low densities and spread over a large geographical area [35]. When the adults are ready to spawn, they migrate to shallow coastal waters. During this migration, they will make extensive vertical movements through the water column, and show a greater association with the sea bed [36]. Males arrive at the coast earlier than females in order to seek out and establish a territory/nesting site. When females arrive, they deposit their eggs in the male's nests, where the male fertilizes the eggs and the females leave quickly [37]. Males guard the eggs that they fertilized until they hatch. Following hatching, the larvae attach to substrates, including seaweed and floating seaweed clumps [38]. Juveniles remain in shallow water areas for approximately 6–12 months before gradually making their way to the feeding grounds offshore. Low recapture rates in tag-recapture studies [39] and limited age classes of spawning lumpfish [40] have led to the suggestion that post-spawning mortality is high and that they have adopted a semelparous life strategy. However, tagging studies have also shown that facultative iteroparity is possible, and that females which return to spawn the following year, spawn at roughly the same time [41] and in the same area [42] as they did pre-viously, in support of homing behavior.

Earlier studies, using 10–11 microsatellite loci, have revealed the existence of three major genetic groups of lumpfish: 1) western Atlantic, including samples from USA, Canada and Greenland, 2) eastern Atlantic, including samples from Iceland and Norway, and 3) the Baltic [43–45]. It was suggested that the major genetic break between western and eastern Atlantic is formed by the cold polar currents going south, keeping populations separated, and that the Baltic Sea population likely became isolated after the colonization of the brackish Baltic Sea around 8,000 years ago [43]. Garcia-Mayoral et al. [44] found latitudinal differentiation along the West coast of Greenland consistent with an isolation-by-distance (IBD) model, and a genetic break between populations in Northwest and Southwest Greenland. In the eastern Atlantic, lumpfish from the English Channel and Iceland have been found to be genetically distant from each other and all other eastern Atlantic samples [45]. There is contradictory evi-dence on whether there is structuring among Norwegian lumpfish [45, 46]. So far, only smaller sets of microsatellite loci have been utilized in population genetic studies of lumpfish across the North Atlantic.

In this study, we aim to improve our understanding of connectivity and population genetic structure of lumpfish across the entire North Atlantic, with a focus on the eastern Atlantic and the Norwegian coast where lumpfish are increasingly being used as cleaner fish, increasing the risk of human mediated gene-flow. In order to achieve this, we first identified a set of genome-wide SNP markers using 2b-RAD sequencing [47]. Thereafter, we selected a set of putatively

discriminatory SNPs (hereon referred to as the targeted SNPs) to resolve regional population structure, to be used in a geographically much more comprehensive study. We genotyped a high number of individuals ($N$ = 1,669), collected over the entire species' distribution (Fig 1), including previously unexplored northernmost areas in the north-eastern Atlantic. Finally, both datasets were analyzed jointly with several environmental variables to identify patterns and drivers of local adaptation.

## 2 Material and methods

### 2.1 Sampling

In total, 1,669 lumpfish collected in the period 2008–2020 from 40 locations were included in this study (Fig 1; Table 1; S2 File). For lumpfish collected over a larger area, the average latitude and longitude were used (Fig 1; Figure 2 in S1 File). Four locations had temporal replicates, which were treated as independent samples to investigate temporal genetic variation. Water-way distances among all sampling sites were determined with the least path method implemented in package *marmap* version 1.0.5 [48] in the R environment, version 4.0.3 [49]. Additional information regarding the ethical, cultural, and scientific considerations specific to inclusivity in global research is included in the S1 Checklist.

### 2.2 DNA isolation, sequencing and data filtering

Genomic DNA was extracted from fin clips stored in absolute ethanol with the Qiagen DNeasy Blood & Tissue Kit following the manufacturer's instructions. Ninety-five fish from 10 locations across the species distribution area (Fig 1; Table 1) were selected for the modified restriction-site-associated DNA sequencing, 2b-RAD [50]. Library preparations were performed as detailed in Faust et al. [33] with the addition of degenerate tags for removing PCR duplicates. Sequencing was done at SciLifeLab in Sweden on an Illumina NovaSeq platform. The resulting raw sequences are available on NCBIs Sequence Read Archive [BioProject PRJNA858951]. PCR duplicates, adapter sequences and low-quality reads were removed, and the remaining sequences were mapped to the *C. lumpus* draft genome [51] with *bowtie2* [52]. SNPs were called with UnifiedGenotyper GATK [53]. Variant score quality was recalibrated (VQSR) using site identity across five technical replicates from Canada, Iceland and Norway as a training set. Sites with more than 10% missing data and a fraction of heterozygotes above 0.5 (possible lumped paralogs) were removed, leaving a total of 7,301 SNPs. After adding a minor allele frequency (MAF) filter of $> 0.1$ in the total dataset, and a minor allele count of at least two, a final dataset of 4,393 SNPs, was retrieved. This genome-wide dataset was used to investigate general global population genetic patterns of lumpfish, as well as to select a subset of discriminatory SNPs.

### 2.3 Selection of SNP markers and genotyping

To find regionally discriminatory markers, non-linked (i.e., not in linkage disequilibrium) SNPs with high divergence ($F_{ST}$) were filtered from multiple pairwise comparisons. SNPs with $F_{ST}$ values above 0.4 were selected from the pairwise comparisons between the North-East Atlantic (Icelandic, UK and Norwegian samples) and 1) Greenland, 2) North America, and 3) the Baltic Sea for designing primers. For separation among the Norwegian and Scottish samples and among all the North-East Atlantic samples, 200 and 100 SNPs with the highest $F_{ST}$ estimates were chosen, respectively (Table A-B in S2 Table). As many SNPs showed high divergence in multiple comparisons, this resulted in a total of 393 SNPs to be used for primer design. $F_{ST}$ for each SNP was calculated with *diveRsity* [54]. Primer design, amplification and

**Table 1. Information of samples ordered from east to west and from north to south.** The coordinates are approximate. Sample codes shown with grey backgrounds are temporal replicates from the same location. Bold codes denote samples with both SNP and genome wide data. Baltic Sea samples were not considered temporal replicates as they were collected from a wider area as part of different surveys. N displays number of individuals after filtering, followed by the number of individuals which did not pass filtering in brackets. Additional metadata can be found in S2 File.

| Location | Code | Region | Coordinates | | N | Collection time | Type* |
|---|---|---|---|---|---|---|---|
| | | | Lat | Lon | | | |
| Barents Sea | BAR_AR | Arctic | 74.760 | 28.774 | 45 (1) | August 2020 | migratory |
| Svalbard, open sea | SVA_AR | Arctic | 75.219 | 9.754 | 34 | September 2020 | migratory, 26 adults, 8 juveniles |
| Isfjorden, Svalbard | ISF_AR | Arctic | 78.394 | 16.593 | 36 | October 2019 | all juveniles |
| Alta, Norway | ALT_NO | Norway | 70.475 | 21.811 | 39 | July 2019 | 34 adults, 5 juveniles |
| Vesterålen, Norway | VES_NO | Norway | 68.992 | 15.218 | 1 (6) | May 2020 | all adults |
| Namsen, Norway | NAM_NO | Norway | 64.482 | 11.291 | 15 | June 2019 | 12 adults, 3 juveniles |
| Flatanger, Norway | **FLA_NO** | Norway | 64.505 | 10.676 | 59 (1) | March 2019 | likely spawning |
| Møre, Norway | MOR_NO | Norway | 63.347 | 8.362 | 23 | May 2019 | 15 adults, 8 juveniles |
| Sognefjorden, Norway | SOF_NO | Norway | 61.266 | 4.845 | 49 (1) | May 2019 | 31 adults, 19 juveniles |
| Hardangerfjorden, Norway | HAR_NO | Norway | 60.060 | 5.143 | 48 (4) | May 2019 | 38 adults, 14 juveniles |
| Boknafjorden, Norway | BOK_NO | Norway | 59.217 | 5.867 | 50 | April 2019 | 21 adults, 29 juveniles |
| Flekkefjorden, Norway | **FLE_NO** | Norway | 58.188 | 6.609 | 70 | March 2019 | likely spawning |
| Skagerrak (ICES SQ 45F9) | SKA_DK | Skagerrak/Kattegat | 58.250 | 9.500 | 57 | July 2019 | migratory |
| Orust, Sweden | ORU_SE | Skagerrak/Kattegat | 58.045 | 11.182 | 81 | 2019 | all juveniles |
| Falkenberg, Sweden | FAL_SE | Skagerrak/Kattegat | 56.462 | 12.308 | 66 (3) | 2019 | all likely adults |
| Svanshall, Sweden | SVAN_SE | Skagerrak/Kattegat | 56.160 | 12.381 | 80 | May 2019 | all likely adults |
| Ålbæk, North Kattegat | ALB1_DK | Skagerrak/Kattegat | 57.590 | 10.420 | 10 | March 2018 | all adults, spawning |
| Ålbæk, North Kattegat | ALB2_DK | Skagerrak/Kattegat | 57.590 | 10.420 | 20 | February 2020 | all adults, spawning |
| Hals, Kattegat | HAL_DK | Skagerrak/Kattegat | 56.970 | 10.330 | 22 | March 2019 | all adults, spawning |
| Helsingør, Øresund | HEL_DK | Skagerrak/Kattegat | 56.060 | 12.593 | 18 (1) | February 2020 | all adults, spawning |
| Baltic Sea, Bornholm Basin | **BAL1_BS** | Baltic Sea | 54.957 | 15.288 | 20 | 2011–2012 | >17cm 17, 12-17cm 3 |
| Baltic Sea, Gotland Basin | BAL2_BS | Baltic Sea | 57.950 | 18.970 | 20 | 2017 | >17cm 10, 12-17cm 8, <12cm 2 |
| Baltic Sea, Bornholm Basin | BAL3_BS | Baltic Sea | 55.700 | 15.200 | 7 | March 2020 | >17cm 4, 12-17cm 3 |
| Baltic Sea, Gotland Basin | BAL4_BS | Baltic Sea | 58.174 | 18.003 | 46 | October 2020 | >17cm 1, 12-17cm 27, <12cm 18 |
| Baltic Sea, Bornholm Basin | BAL5_BS | Baltic Sea | 55.607 | 15.772 | 4 | June 2020 | >17cm 4 |
| North Sea, mid sea | NOR_NS | North Sea/British Isles | 59.020 | 0.462 | 35 | August 2019 | migratory |
| English Channel, British Isles | **ECH_NS** | North Sea/British Isles | 50.042 | -2.522 | 20 | 2015 | adults, 9 males, 11 females |
| Outer Hebrides, British Isles | **OHE_NS** | North Sea/British Isles | 58.160 | -6.315 | 18 (2) | 2017 | all adults |
| Vopnafjörður, Iceland | VOP_IS | Iceland | 65.777 | -14.702 | 60 | April 2019 | all spawning females |
| Bakkafjörður, Iceland | BAK_IS | Iceland | 66.166 | -14.991 | 50 | April 2011 | all spawning females, 3–5 years |
| Skagastrønd, Iceland | SKA1_IS | Iceland | 65.781 | -20.461 | 50 | February 2012 | spawning adults, all males |
| Skagastrønd, Iceland | SKA2_IS | Iceland | 65.781 | -20.461 | 79 (1) | April 2019 | spawning adults, all females |
| Bolungarvik, Iceland | BOL_IS | Iceland | 66.158 | -23.225 | 49 (1) | May 2012 | spawning, 48 females, 2 males |
| Breiðafjörður, Iceland | **BRE_IS** | Iceland | 65.235 | -23.291 | 49 | June 2011 | adults, 4 males, 42 females, 3 NA |
| Sandgerði, Iceland | SAN_IS | Iceland | 64.037 | -22.773 | 49 | May 2012 | spawning, 48 females, 1 male |
| Upernavik, Greenland | **UPE_GR** | Greenland | 72.790 | -56.151 | 10 | June 2014 | spawning |
| Qeqertarsuaq, Greenland | QEQ_GR | Greenland | 69.221 | -53.698 | 17 (8) | June 2014 | spawning |
| Nuuk, Greenland | NUU1_GR | Greenland | 64.172 | -51.764 | 7 (7) | 2011 | all adults, 6 females, 8 males |
| Nuuk, Greenland | **NUU2_GR** | Greenland | 64.172 | -51.764 | 20 | June 2014 | spawning |
| Qaqortoq, Greenland (South) | QAQ_GR | Greenland | 60.737 | -45.965 | 19 (6) | April 2015 | spawning |
| Newfoundland, Canada | **NEW_CA** | North America | 49.943 | -63.910 | 20 | June 2008 | all spawning females |
| Gulf of St. Lawrence, Canada | GSL_CA | North America | 49.429 | -59.153 | 88 (1) | 2019 | 54 adults, 35 juveniles |
| Maine (Scantum Basin), USA | **MAI1_US** | North America | 42.833 | -70.580 | 19 (1) | 2012 | 9 females and 6 males, 3 juveniles, 2 NA |
| Maine (Scantum Basin), USA | MAI2_US | North America | 42.833 | -70.580 | 18 (1) | 2019 | all adults, 13 females, 5 males, 1 NA |

*Fish from outside the Baltic Sea ≤330g and/or ≤20 cm was considered as juveniles. When sex, spawning status and/or age are known, they are listed. In the Baltic Sea, fish are divided into three size classes.

genotype calling were done using the Agena MassARRAY iPLEX Platform as described by Gabriel et al. [55]. The final selected set of markers consisted of 198 SNPs divided into seven assay groups (Table A-B in S2 Table).

All 1,669 lumpfish along with 19 technical replicates (i.e., replicate individuals) were genotyped with the selected 198 SNPs. Each 384-plate contained two to four negative controls. Technical replicates were used to estimate genotyping consistency. A total of 59 loci did not amplify or produce a clear clustering pattern and were discarded (Table A in S2 Table). Individual and locus-wise call rates were checked with R package *dartR* [56], and loci with ≥15% missing data ($N = 4$) and individuals with ≥ 20% missing data ($N = 44$), were discarded. In Vesterålen, a single fish passed the filtering steps, and this sampling site was removed from further analysis. The final dataset consisted thus of 1,597 lumpfish from 39 different sites, genotyped with 139 informative SNPs. Technical replicates did not show discrepancies besides a likely drop-out detected for one individual in three loci. The final targeted SNP dataset had a total of 2.34% missing data. The selected SNPs were distributed across all 25 chromosomes in the lumpfish genome (Table A in S2 Table), and many of them were located within ($N = 64$) or in the immediate vicinity (less than 2kb away; $N = 16$) of known/annotated gene sequences.

## 2.4 Population genetic analysis

The genome-wide, sequence-derived dataset consisting of 95 fish from 10 locations and 4,393 SNPs, is hereafter referred to as the "genome-wide dataset" (S3 File). The genotype data from the final 139 SNP panel is referred to as the "targeted dataset" (S4 File). All analyses were conducted with both datasets unless otherwise stated. Besides analyzing samples, several analyses were carried out using larger pooled groups based on geographic regions (as given in Table 1). When applicable, grouping was made differently based on results received from the upstream analyses.

**2.4.1 Genetic variation and its division.** Expected and observed heterozygosity, allelic richness, and inbreeding coefficient ($F_{IS}$) were estimated with the *diveRsity* package [54] in R. Confidence intervals at 95% were calculated for the $F_{IS}$ indices using 1,000 bootstraps. Deviation from Hardy-Weinberg expectations (HWE) was estimated both locus- and sample-wise with Fisher's exact probability test and complete enumeration using Genepop v.1.1.7 [57]. For the targeted dataset, the distribution of variation between geographic regions, between sampling sites (and their temporal replicates) within regions, and within samples was investigated with AMOVA in the R package poppr v.2.9.1 [58]. Statistical significance for the variance components was obtained with 999 permutations using the ade4 package, v.1.7–16 [59]. Pairwise $F_{ST}$-values [60] were calculated with StAMPP [61] in R. Corresponding 95% confidence intervals and p values were calculated with 2,000 bootstraps, and corrected with False discovery rate (FDR) [62] for multiple comparisons. Pairwise $F_{ST}$ were visualized with UPGMA dendrogram with *as.dist()* and *hclust()* functions in 'stats' package in R.

**2.4.2 Individual-based clustering.** Three clustering approaches were employed to investigate and visualize the genetic differentiation among individuals. First, clustering was done with Principal Component Analysis (PCA) with the R package *ade4* [59, 63]. PCA is a multivariate exploratory approach without assumptions on populations or their boundaries. Next, the variation that maximized among-group differences was identified using *adegenets'* v.2.1.3 [64, 65] discriminant analysis of principal components (DAPC) with the *xvalDapc* function to choose the optimal number of principal components. The third method, STRUCTURE v.2.3.4 [66, 67], is a model-based Bayesian clustering method that uses a predefined number of *K* clusters to estimate the posterior probability of each individual's genotype to originate from each cluster. All STRUCTURE runs were performed using the default admixture model with

correlated allele frequencies and with location information given a priori [68]. A total of 50 000 MCMC (Markov Chain Monte Carlo) repetitions were used in each run after 20 000 repeats were discarded as burn-in. *K* was set from 1 to 10 or 12 (depending if whole or partial dataset was used, see below), and the number of iterations was set to 5. To determine the optimal *K*, bar plots were inspected visually and runs analyzed with the StructureSelector software [69]. The software summarizes results as the optimal Ln Pr(X|K) given by the STRUCTURE software and the ad hoc summary statistic ΔK [70], which identifies the uppermost level of population hierarchy. Moreover, the StructureSelector calculates MedMed, MedMean, MaxMed and MaxMean as Puechmaille [71] described. Results from the runs for the different values of K were averaged with CLUMPAK [72] using the LargeKGreedy algorithm and 2,000 repeats.

As higher levels of structure can effectively mask lower levels of structure [71] a hierarchical approach was employed. The above-mentioned clustering analyses were therefore performed first for the whole dataset, followed by a separate analysis for sampling sites that were geographically close and showed similar admixture profiles. For each of the investigated regions, the most informative loci behind the observed structure were determined with the help of PCA loadings. The threshold for being informative was set to 0.1.

**2.4.3 Population graphs.** Genetic structure for the targeted SNP data was also analyzed with Population Graphs, a graph-theoretic approach that uses conditional genetic distance [73] as the response variable, which is primarily created by both gene-flow and shared ancestry. This approach has been shown to capture underlying demographic processes more accurately than methods based on pairwise estimates of genetic structure or various genetic distance metrics [74]. The identification of genetic covariance structure among populations is independent of evolutionary assumptions aiming to minimize Hardy–Weinberg and linkage disequilibrium within populations [73]. Population Graphs were constructed using the R packages *popgraph* and *gstudio* [75, 76] and topological analyses were performed using the *igraph* package [77].

For Population Graph, graph distance was estimated as the minimal topological distance connecting pairs of sampling sites (nodes) where the distances between nodes in the network, based on genetic covariance, was evaluated in relation to physical separation of nodes on the landscape under a model of isolation by graph distance, IBGD [73]. A relatively small graph distance between spatially distant sites indicates long-distance gene-flow (extended edges). Conversely, geographical or ecological barriers that impede gene-flow relative to other localities with similar distances generate relatively high graph distances (compressed edges). Correlations and detection of extended and compressed edges were determined by regressing geographic and graph distances. Analyses were conducted using 35 geographically-explicit locations after pooling temporal samples, and pooling all Baltic Sea samples in one.

**2.4.4 Environmental association analyses.** Environmental factors deemed likely to be important to lumpfish populations were filtered from the Bio-ORACLE database (https://www.bio-oracle.org; [78, 79]) for the sampling sites (Table A in S1 Table; Figure 3 in S1 File). The decision of which factors to include was primarily based on expert opinion (C. Dürif, personal communication) but also comprised of additional factors commonly included in similar analyses of other marine fish species. Collinearity between variable pairs was investigated with the R package *corrr* v.0.4.3 [80] and *corrplot* version 0.84 [81]. Highly correlated variables were merged into new synthetic variables (Table B in S1 Table) using a hierarchical clustering method in the R package *ClustOfVar* v.1.1 [82] and 100 bootstraps were used to estimate the obtained grouping stability.

To detect signals of selection and adaptation to the local environment, datasets were analyzed with two independent methods (Table B in S1 Table). First, latent factor mixed model

(LFMM), a univariate association method between genotypic and environmental variable matrices was run using functions implemented in the R package *LEA*, v.3.2.0 [83]. Here, the optimal number of principal components explaining the genetic variation (*K*) was determined with the function *pca()* and by choosing the "knee" point in the associated scree plot. To get the required complete genotypes, missing data was imputed using the *snmf()* function with the selected *K*, default settings and 10 repetitions. Best run of K clusters was determined as the one having the lowest cross entropy. Then, using the imputed dataset, the *lfmm2()* and *lfmm2.test()* functions were used to estimate latent factors based on the exact least-square approach, and their adjusted *p* values. Here, the default value was used for the lambda parameter, and a linear model was selected for the following *p* value estimation. For the obtained *p* values between each environmental variable and SNP, corresponding *q* value was calculated using the R package *qvalue*, v.2.22.0 [84]. The threshold for false discovery rate was set to 5%.

Further, the R package, *pcadapt* v.4.1.0 [85], was used for the genome-wide data to perform a SNP outlier scan to cross-validate the obtained results from the LFMM analysis. *pcadapt* is a non-constrained ordination method that like LFMM uses a PCA approach to define the underlying population structure prior to the outlier scan. The test for outliers is based on the correlations between genetic variation and the first *K* principal components. Same *K* as for the LFMM approach was used here, and the *q* value threshold was set to 5%.

Finally, for the targeted dataset, redundancy analysis (RDA; see e.g. [86]) was adopted. RDA is a linear multivariate ordination method that can simultaneously analyze multiple environmental variables and genetic markers. Here, the geographic distance matrix between all sampling sites was first transformed into corresponding dbMEMs (distance-based Moran's Eigenvector Maps) using *adespatial*, v.0.3–14 [87]. Significant geographic and environmental variables, explaining allele frequency variation across samples, were identified using the *forward.sel()* function. Only significant variables were included in the actual RDA analysis with the R package *vegan*, v.2.5–7 [88]. RDA was performed using the full model for regression, followed by a partial model where geographic location was controlled for. Variance inflation factor was checked to be below the recommended 10 for all variables, and permutation tests were repeated 1000 times.

## 3 Results

### 3.1 Genetic variation and its division

Expected heterozygosity over the 4,393 SNPs varied between 0.124 and 0.151 across samples (S3 Table). Observed heterozygosity estimates were consistently slightly higher between 0.132–0.179. Heterozygosity measured with the targeted 139 SNP panel was higher, between 0.193 and 0.280 ($H_e$). In 13 samples, observed heterozygosity deviated significantly from expected heterozygosity, and $F_{IS}$ was positive in nine of them.

Most samples were significantly differentiated from each other (Table 2), and all studied levels of population hierarchies displayed significant divisions (Table 3). None of the temporal replicates were significantly differentiated (S4 Table). Global mean $F_{ST}$ was 0.094 (SD ± 0.057) for the genome-wide dataset, and slightly higher, 0.115 (SD ± 0.007) for the targeted dataset, as expected for outlier loci. Three major genetic groups were clearly identified: The highly-differentiated West and East Atlantic ($F_{ST\ (genome-wide)}$ 0.087–0.154), and the Baltic Sea ($F_{ST\ (genome-wide)}$ >0.08 and >0.16 against West and East Atlantic, respectively). Estimates of genetic divergence among these three groups were consistently higher, about two to five-fold, for the targeted dataset (Table 2, Table B in S4 Table).

In addition to the three major branches identified, the targeted SNP dataset also showed separation between North America (USA and Canada) and West Greenland/Svalbard-

**Table 2. Pairwise $F_{ST}$ values among samples of lumpfish (*Cyclopterus lumpus*) with both genome-wide and targeted SNPs.** Below diagonal shows estimated $F_{ST}$s. The first value is from the genome-wide data (4,393 SNPs) and in parentheses from the targeted data (139 SNPs). Above diagonal shows corresponding p values based on 2,000 bootstraps. If only one value is given, both comparisons coincided with that value. Pairwise $F_{ST}$ table including remaining samples from the targeted SNP data is given in Table A in S4 Table.

| | Norway | | Baltic Sea | British Isles | | Iceland | Greenland | | North America | |
|---|---|---|---|---|---|---|---|---|---|---|
| | Flatanger | Flekkefjord | Bornholm | English Channel | Outer Hebrides | Breiðafjörður | Upernavik | Nuuk | Newfoundland | Maine |
| | FLA_NO | FLE_NO | BAL1_BS | ECH_NS | OHE_NS | BRE_IS | UPE_GR | NUU2_GR | NEW_CA | MAI1_US |
| Flatanger | | 0.010 (0) | 0 | 0 | 0.027 (0) | 0 | 0 | 0 | 0 | 0 |
| Flekkefjord | 0.004 (0.007) | | 0 | 0 | 0.007 (0) | 0 | 0 | 0 | 0 | 0 |
| Bornholm | 0.083 (0.173) | 0.082 (0.162) | | 0 | 0 | 0 | 0 | 0 | 0 | 0 |
| English Channel | 0.025 (0.063) | 0.024 (0.048) | 0.099 (0.220) | | 0 | 0 | 0 | 0 | 0 | 0 |
| Outer Hebrides | 0.003 (0.016) | 0.004 (0.022) | 0.087 (0.198) | 0.021 (0.055) | | 0 | 0 | 0 | 0 | 0 |
| Breiðafjörður | 0.013 (0.036) | 0.015 (0.046) | 0.089 (0.176) | 0.035 (0.111) | 0.012 (0.054) | | 0 | 0 | 0 | 0 |
| Upernavik | 0.106 (0.251) | 0.113 (0.251) | 0.179 (0.313) | 0.134 (0.309) | 0.112 (0.272) | 0.099 (0.223) | | 0 | 0 | 0 |
| Nuuk | 0.103 (0.216) | 0.105 (0.214) | 0.160 (0.278) | 0.124 (0.270) | 0.104 (0.237) | 0.087 (0.171) | 0.021 (0.037) | | 0 | 0 |
| Newfoundland | 0.150 (0.269) | 0.154 (0.264) | 0.204 (0.309) | 0.170 (0.314) | 0.152 (0.291) | 0.141 (0.235) | 0.060 (0.106) | 0.065 (0.092) | | 0 |
| Maine | 0.140 (0.216) | 0.144 (0.219) | 0.196 (0.282) | 0.161 (0.274) | 0.141 (0.232) | 0.132 (0.186) | 0.076 (0.107) | 0.078 (0.099) | 0.026 (0.032) | |

All p values remain significant (p<0.05) after correction for multiple testing.

Isfjorden. Also, Kattegat in the Baltic transition zone, the English Channel, Iceland and three fjords in southwestern Norway showed separation from their neighboring samples (Fig 2). In the West Atlantic, US and Canadian samples differed from Greenland ($F_{ST} \geq 0.07$) and from each other with a genome-wide mean $F_{ST}$ of ~ 0.026. Within Greenland, samples north and south of 65˚N were genetically different ($F_{ST} \geq 0.021$).

In the Arctic, Isfjorden stood out as highly differentiated from the samples collected off-shore from Svalbard and the Barents Sea area, as well as from northern Norway and Iceland (Table A in S4 Table). Instead, Isfjorden shared a close resemblance with western Greenland (Fig 2). Lumpfish from the Baltic Sea formed a highly distinct genetic group. Fish collected in the English Channel also formed a distinct group. Among the remaining East Atlantic samples, lumpfish collected from the Kattegat area, Iceland, and southwestern Norwegian fjords (Sognefjorden, Hardangerfjorden and Boknafjorden) each formed separate clusters. Lumpfish from the rest of Norway, Outer Hebrides, North Sea, Skagerrak, and open Arctic waters

**Table 3. Distribution of genetic variance of lumpfish (*Cyclopterus lumpus*) for 139 targeted SNPs.** Samples are equal to sets of individuals collected at the same site at the same time point, and the regions represent the geographic proximity of sampling sites, as described in Table 1.

| Variance source | d.f. | Sum of squares | Variance components | % of variation | p value |
|---|---|---|---|---|---|
| Among regions | 7 | 12 122 | 4.3 | 10.4 | <0.001 |
| Among samples within regions | 35 | 3 659 | 0.9 | 2.2 | <0.001 |
| Among individuals within samples | 1 554 | 58 735 | 1.5 | 3.6 | <0.001 |
| Within individuals | 1 597 | 55 634 | 34.8 | 83.8 | |

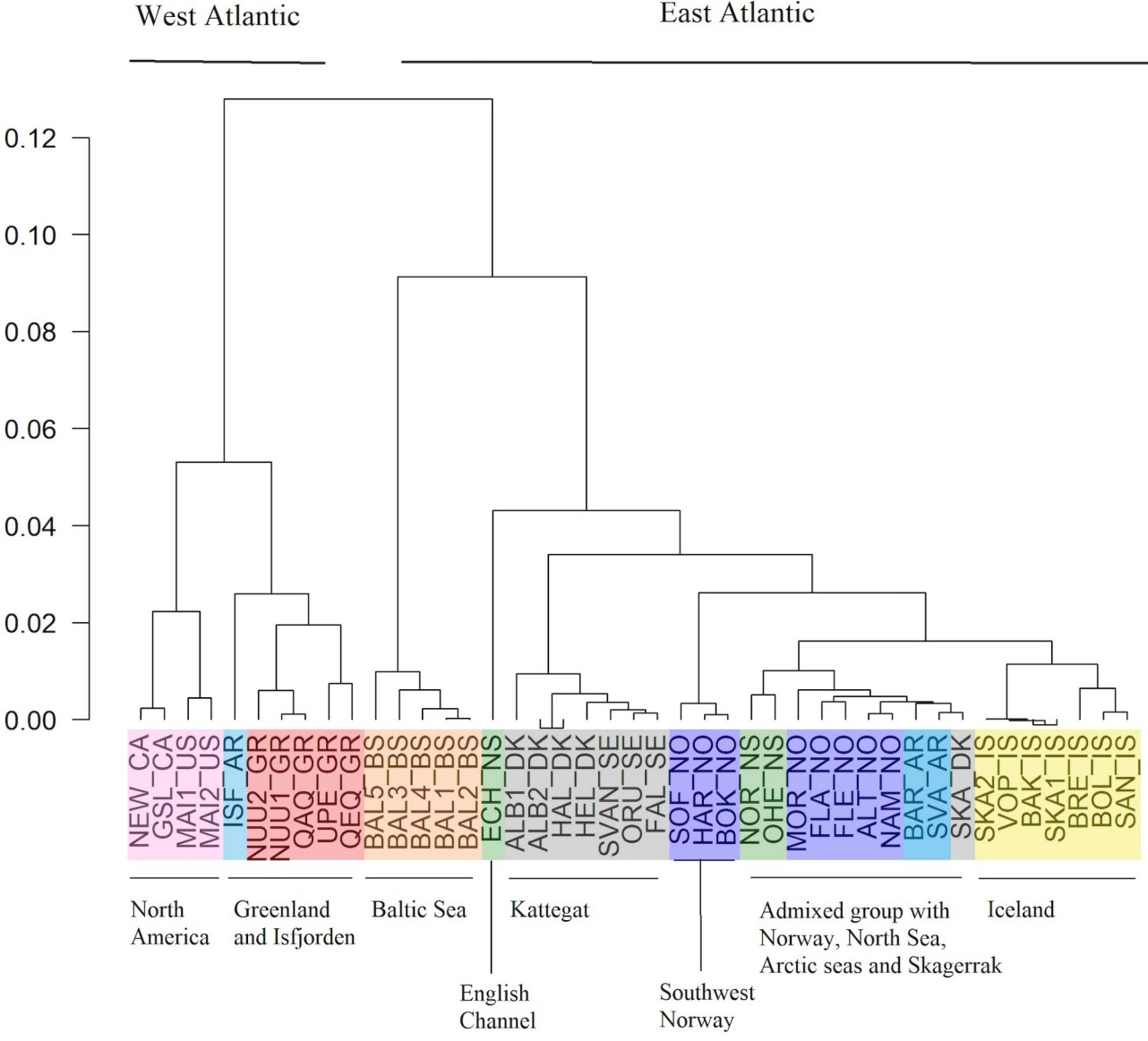

**Fig 2. Dendrogram of population pairwise $F_{ST}$ estimated in lumpfish (*Cyclopterus lumpus*) based on a targeted dataset with 139 SNPs.** Sample codes and colors indicate geographic regions as described in Table 1.

(Svalbard and Barents Sea) formed one group with relatively low genetic divergence among samples (Fig 2), referred to as the East Atlantic Group below.

## 3.2 Individual-based clustering

Plotting population structure with PCA revealed similar large-scale division as pairwise $F_{ST}$ estimates (S5 File). The first axis divided the East and West Atlantic in both genome-wide and targeted datasets, and explained 11.1% and 10.9% of the total variation respectively. The second axis separated the Baltic Sea samples from the rest and explained 3.45% and 4.60% respectively. Using the DAPC approach and regional partitioning of the data showed similar results as the PCA but with clearer clustering (S5 File). Lumpfish from the USA and Canada formed separate clusters, as did fish from Greenland, with a North-South separation. Icelandic

lumpfish showed a West-East separation. In mainland Norway, three overlapping clusters were formed, roughly separating northernmost Alta, from large fjords in southwestern Norway (Sognafjorden, Hardangerfjorden and Boknafjorden), and from mid- and south-Norway (Flatanger and Flekkefjord). The Baltic Sea and the North Sea transition zone, Skagerrak and Kattegat, separated in three distinct clusters, with the exception of one likely migrant from the Baltic in the Skagerrak offshore sample. Within the East Atlantic group, weak separation could be seen between northernmost areas (Barents Sea, Svalbard and Alta), mainland Norway (Flatanger and Flekkefjord), and Skagerrak-North Sea (including Outer Hebrides).

The results from STRUCTURE were consistent with those from the PCA/DAPC approaches but provided additional insights. In the genome-wide dataset, the division between western and eastern Atlantic was clear, but many individuals also shared signs of admixture (Fig 3). In the East Atlantic, all lumpfish shared at least ~ 50% of their genetic information. $K = 7$ was deemed as the best fit for the data (S5 Table), as this was the highest value creating visually discrete clusters, corresponding to six genetic clusters: North America, Greenland, Iceland, English Channel, Baltic Sea, and finally Norway and the Outer Hebrides which clustered together.

The targeted SNP dataset revealed nine major clusters ($K = 9$), showing an additional cluster in Kattegat, the Norwegian southwestern fjords and an east-west divergence across Iceland. Hierarchical clustering analysis of West Atlantic samples showed a clear separation between US and Canada, and between northwest and southwest Greenland (Fig 4; S5 Table). In the samples from USA-Maine, a possible migrant from Canada was observed together with a possible first-generation hybrid (also clearly visible in DAPC plot; S5 File).

The Baltic Sea and the North Sea transition zone, Kattegat and Skagerrak, were separated into three clusters (Fig 4; S5 Table). One individual caught in Skagerrak was clearly a migrant of Baltic Sea origin, as also seen in the PCA and DAPC (S5 File). The northernmost samples from Kattegat (Orust) displayed a high proportion of assignments to Skagerrak. Moreover, individuals from western and eastern Kattegat formed separate clusters. Among lumpfish the Norwegian mainland fjords, two distinct genetic clusters could be observed (Fig 4; S5 Table). As seen in both PCA and DAPC analysis, several individuals from the large southwestern Norwegian fjords (Sognefjorden, Hardangerfjorden and Boknafjorden) clustered together,

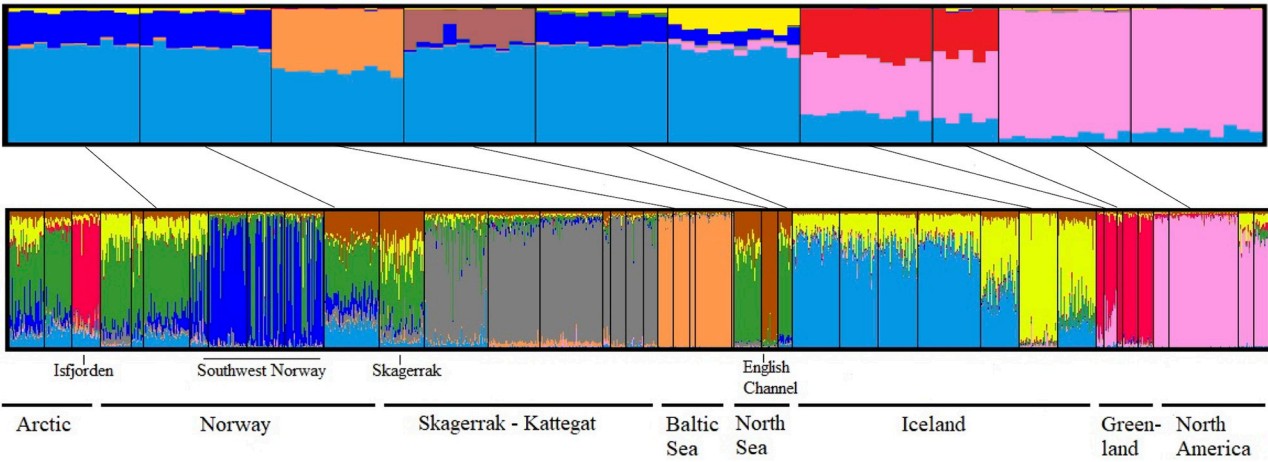

**Fig 3. Assignment of lumpfish (*Cyclopterus lumpus*) into genetic K clusters for the genome-wide dataset with 4,393 SNPs (K = 7) in the top panel, and targeted dataset with 139 SNPs (K = 9) in the bottom panel.** Each bar is one individual and the different colors represent the proportional assignment to the different K clusters. Samples that were genotyped with both methods are connected with black lines between the top and bottom panel.

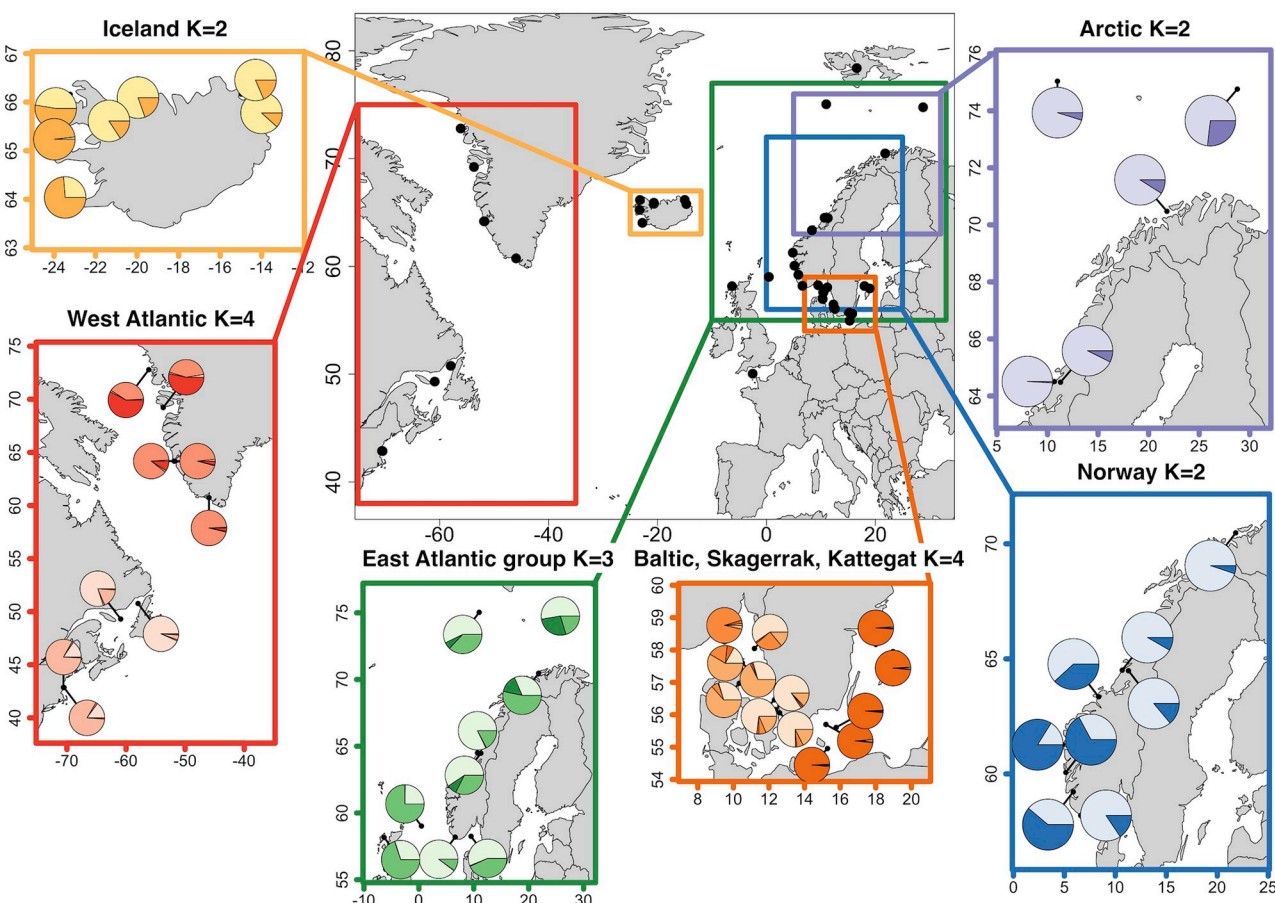

**Fig 4. Regional STRUCTURE assignment of lumpfish (*Cyclopterus lumpus*) based on targeted 139 SNPs.** Square represents separate structure runs for different regions where a set of samples, which due to their genetic and geographic closeness were analyzed together. Each pie chart shows one sample's assignment to the selected number of clusters (*K*) averaged across individuals. Note that some of the samples are included in multiple assignments but that Isfjorden and the English Channel samples were not included in any regional structure analysis due to their uniqueness.

separating them from all other Norwegian lumpfish. Additional levels of genetic structuring separated the northernmost Norwegian samples to some degree. When only samples from the Arctic (open sea and northernmost Norwegian fjords) were considered, *K* = 2 was supported, and an additional Arctic cluster within the Barents Sea was revealed (Fig 4; S5 Table).

### 3.3 Patterns of population connectivity

The distribution of spatial genetic structure in the targeted dataset suggested restricted gene-flow under a model of isolation by graph distance, that is, distances between nodes in the network of genetic covariance in relation to geographic distance between samples (IBGD; Spearman's rho = 0.733, $p < 0.2$ e-16; Figure 4 in S1 File). *Popgraph* depicted a complex web where the 35 nodes (samples) were connected by 119 edges (out of the 595 possible ones; Fig 5; S6 Table). Out of the 119 edges, 27 were proportional, 38 were extended (i.e., more gene-flow than what would be expected given geographic distance), and 54 were compressed (i.e., less gene-flow than what would be expected given geographic distance). The majority, 83% of the compressed edges, indicated less gene-flow occurred within regions. Conversely, 87% of the extended edges connected different regions, and 11% of the extended edges occurred between Norwegian nodes in a range of 700–1500 km of geographic distance.

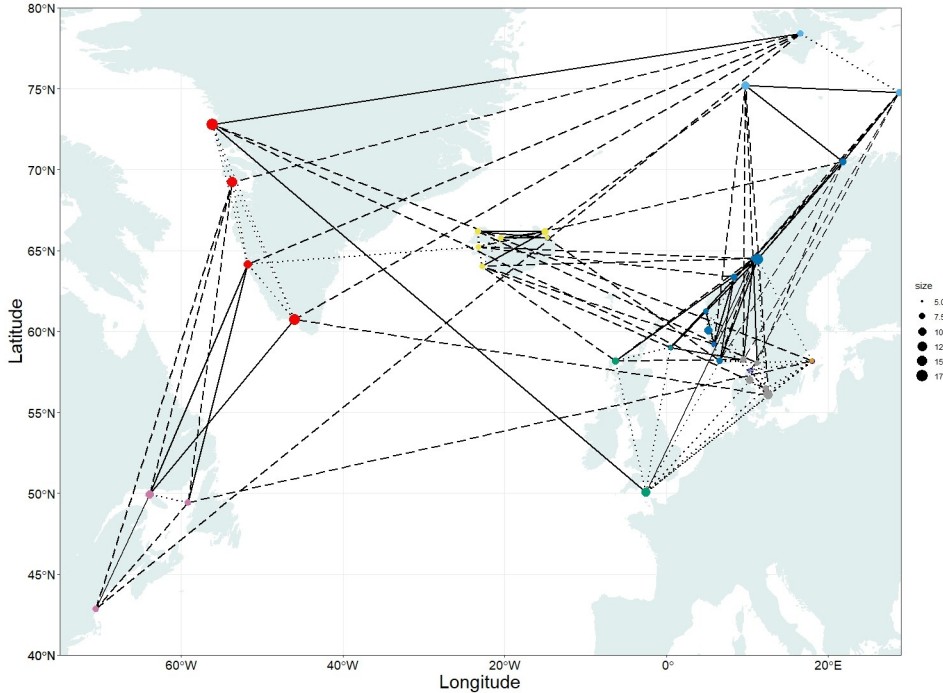

**Fig 5. Population graph representing the distribution of genetic covariance in the global lumpfish (*Cyclopterus lumpus*) population based on 139 SNPs.** Nodes represent samples, and their relative sizes the amount of within-population genetic variance. Edges connecting the nodes represent the pattern of genetic covariance among populations: Dashed lines denote extended edges (i.e., more gene-flow than what would be expected given geographic distance), dotted lines compressed edges, (i.e., less gene-flow than what would be expected given geographic distance), and continuous line proportional edges (i.e., not significantly more or less gene-flow). Colors indicate geographic regions as described in Table 1.

## 3.4 Outlier loci detection and association with environment

Based on the correlation analysis among the 14 environmental factors, five independent synthetic variables (Productivity, Oxygen_MinTemp, Velocity, Temp_max_mean, and Salinity_-Temp_range; Table B in S1 Table) were created for each sample and used further (see S6 File for details). Using LFMM, we identified 58 significant SNP outlier loci–environment associations, in the genome-wide SNP data (Table A-B in S7 Table). PCadapt detected a total of 24 outlier loci out of which nine were shared between the PCadapt and LFMM (Table C in S7 Table). The identified outliers were distributed throughout the 25 lumpfish chromosomes. Most of the outlier loci were located within or in immediate vicinity of annotated gene sequences (Table B-C in S7 Table). For the targeted dataset, LFMM identified 20 significant associations for 15 different SNPs (S8 Table; S6 File). Five of the SNPs were located within genes, two of which were also outliers in the genome-wide dataset.

Finally, based on redundancy analysis, four canonical axes and two synthetic environmental variables (Salinity/Temperature range, Oxygen level/Minimum Temperature) were significantly associated ($p < 0.001$) with the observed patterns of genetic structure. Together they explained 55.5% of the total variation (Fig 6). Partial analysis removing the effect of geography still showed a significant association ($R^2 = 0.086$, $p < 0.001$), albeit much weaker. Outliers were defined as being at least 2.5 standard deviations away from the mean SNP loadings along the significant axes. Only a single SNP (Lump-165) passed that criterion. The same SNP was denoted as a putative outlier in the LFMM analysis with both datasets (Table B in S7 Table; S8 Table).

## Seascape redundancy analysis with SNPs

$R^2 = 0.555$

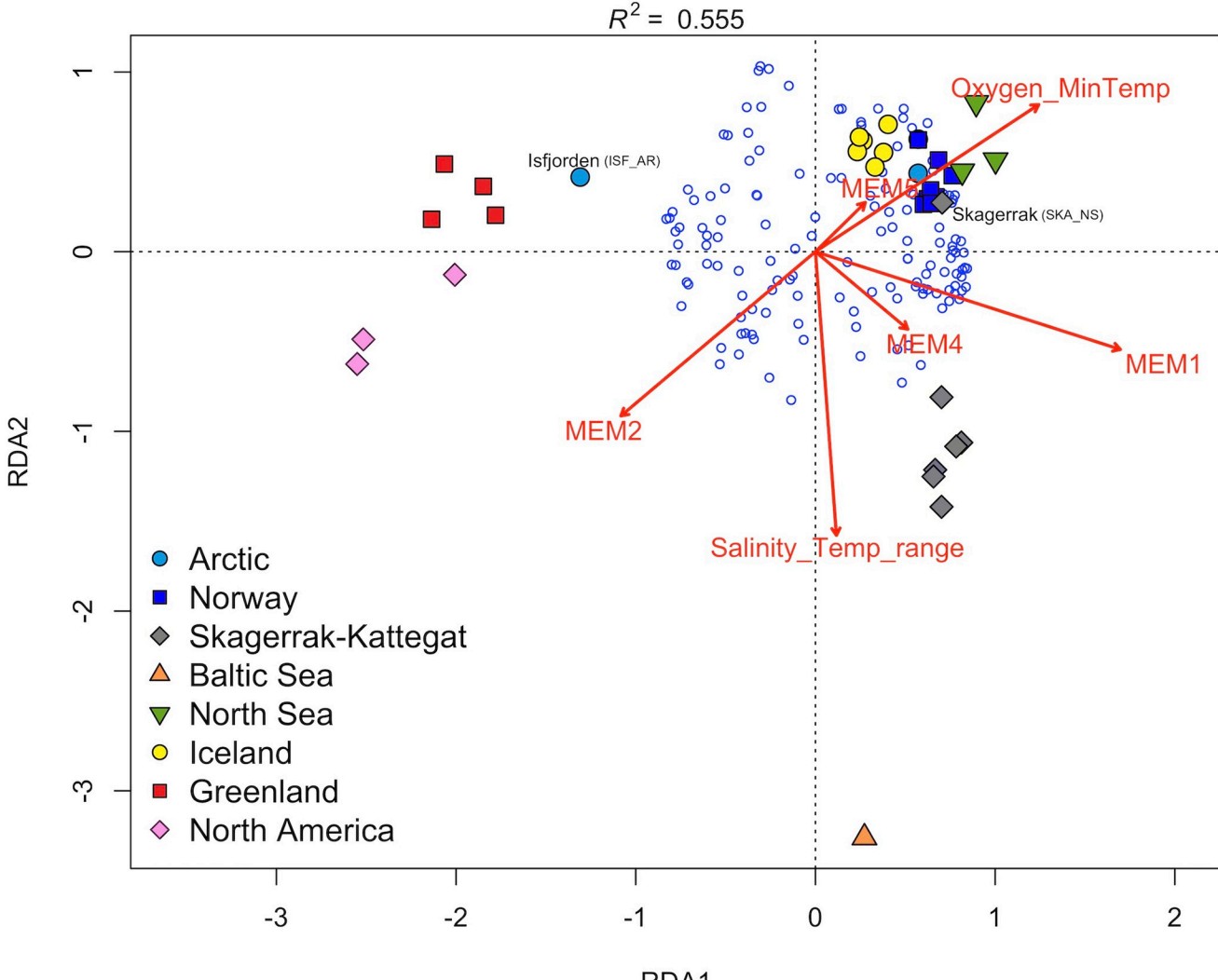

**Fig 6. Seascape redundancy analysis of lumpfish (*Cyclopterus lumpus*).** Moran Eigenvector's Maps (MEMs) decompose spatial relationships among samples based on a spatial weighting matrix. Four MEMs, and two synthetic environmental variables (Salinity/Temperature range, Oxygen level/Minimum Temperature) were significantly associated ($p < 0.001$) with the observed genetic structure. Together they explained 55.5% of the total variation. Samples are color-coded based on their geographic region as described in Table 1, with all Baltic Sea samples pooled as one. Isfjorden and Skagerrak samples are labeled as separated from other samples in their region. Small blue circles denote 139 SNPs.

## 4 Discussion

The patterns of population genetic structure that we observe today result from a complex interplay of evolutionary forces from both the past and present. For lumpfish, strong oceanic currents, demographic history, homing behavior, glacial refugia, and local adaptation have all been suggested as likely drivers of genetic population structure [43–45]. Based on our study, lumpfish populations display extensive global, regional, as well as local patterns of population genetic structure, which are created by different processes and partly mediated by different parts of the genome. Combining genome-wide SNPs with a targeted SNP panel and extensive sampling, we uncovered multiple layers of population genetic structuring across the

distribution of lumpfish. Complex patterns of connectivity among and within samples, and an overall genetic structure that was significantly linked with environmental variability, suggest that populations are locally adapted. For a widely distributed marine fish that undertakes off-shore feeding migrations, natal homing is a likely mechanism for finer-scale genetic differentiation and evolution of local adaptation [89].

## 4.1 Global population structure

The two datasets, i.e., the 4,393 genome-wide SNPs and the targeted panel of 139 SNPs, revealed similar patterns for lumpfish concerning higher-level population structure: We observed clear genetic splits dividing West and East Atlantic, the Baltic Sea, Iceland and the English Channel. These large-scale patterns are concordant with the results from previous population genetic studies [43–45], and likely represent population patterns shaped by demographic history and strong oceanic currents. Similar deep genetic splits across the Atlantic Ocean have been described in other transatlantic species (e.g., Atlantic cod (*Gadus morhua*) [90] and red alga (*Palmaria palmata*) [91]). The split of these species likely occurred >100 000 years ago during the interglacial period, when shallow ocean depths on continental shelves favored transcontinental movement and population expansions.

We found that the global population subdivision of lumpfish based on the genome-wide data alone would best be represented by six groups: North America, Western Greenland, Iceland, English Channel, Baltic Sea, and Norway, together with the Outer Hebrides. While using the targeted dataset and applying a hierarchical approach, we identified additional structuring with a total of nine major genetic groups globally: North America, Greenland together with Isfjorden in Svalbard, eastern and western Iceland split in two, English Channel, Baltic Sea, Kattegat in the North Sea transition zone, fjords in southwestern Norway and finally, an East Atlantic group covering a larger geographic area and including samples from Norway, Arctic open waters, Skagerrak, North Sea and the Outer Hebrides.

## 4.2 Regional population structures

We detected different degrees of sub-structuring within the large regional groups. Some finer-scale population patterns for lumpfish have already been described in previous studies where isolation-by-distance was detected in West Greenland [44], and also suggested in Norway where Averøy in mid-Norway was found to be different from other Norwegian samples [45]. Here we found further structuring within regions that have not been previously described. When we examined the genetic results jointly with geographic location and information on life history stages (Table 1), we found that samples with only breeding and juvenile lumpfish formed regional and clear-cut groups, whereas migrating fish, mid-sea samples and samples known to contain both juveniles and adults were more admixed. Genetic divergence of regional breeding populations is thus in line with observations which suggest homing in lumpfish [41, 42]. Homing creates reproductive barriers and can facilitate the development and maintenance of local adaptation, resulting in adaptive differences between populations [e.g., 92]. This allows for the genetic characteristics that are beneficial for survival and reproduction in that specific environment to be passed down to future generations, leading to the population becoming increasingly well-adapted to that environment over time.

**4.2.1 West Atlantic.** In the western Atlantic, we found significant genetic differentiation among samples collected from Greenland, Canada and USA, as well as a north-south division along Greenland's west-coast, as described previously [43, 44]. Furthermore, the targeted SNP panel produced distinct clustering patterns for the groups (Fig 4; S5 Table), suggesting good regional resolution, and could likely be used to e.g., detect migrants. Indeed, two individuals

collected in Maine, USA, had high admixture proportions (~50 and ~80%) and resembled the Canadian sample, suggesting that they could be migrants or hybrids between the two populations.

**4.2.2 Isfjorden.** Despite being geographically closer to several samples from the East Atlantic, the sample from Isfjorden in Svalbard clustered closely with samples from western Greenland. As this sample was highly divergent from off-shore Svalbard samples, and we did not detect any individuals with similar genetic profile in any nearby samples, this fjord population may represent a regionally divergent gene pool, perhaps specially adapted to its Arctic environment [93]. Svalbard is surrounded by strong Arctic currents that run north and west towards the Arctic Ocean and the Greenland Sea (Figure 6 in S1 File). However, the closest sample in Greenland in this study was over 3,000 kilometers away (along the shortest ice-free waterway). Either Isfjorden is an isolated relic of past gene-flow or a manifestation of still on-going connectivity. It is unknown how far north lumpfish reach, but historical Arctic connectivity could explain the observed pattern. Strong cold polar currents have been proposed as the barrier separating the West and East Atlantic [43], Greenland and Canada, as well as Greenland and Iceland lumpfish populations [44]. These divisions align with our results, which showed significant divergence and no likely migrants across these borders. In order to better disentangle the different scenarios of past connectivity and/or local adaptation, a higher number of genetic markers are needed to investigate if past changes in population size and demographic history can explain the patterns observed in Isfjorden.

**4.2.3 Iceland.** We found a weak but statistically significant east-west subdivision across the Westfjords in Iceland. Breiðafjörður appeared most distinct, whereas fish from Sandgerði and Bolungarvik were more admixed (Fig 4). This pattern was driven by only three outlier loci (Table C in S2 Table), all of them in chromosome 13, and thus possibly suggesting a structural variant or a genomic region for local selection. A region in this chromosome, close to the outliers reported here, was recently discovered to be highly associated with sex [94], indicating that the observed structure could be related to differences between males and females. Repeated sampling of fish of opposite sex from the same location, Skagastrønd (SKA1_IS and SKA2_IS), did not show any significant differentiation, however ($F_{ST}$ = 0.0007, NS). This suggests that sex is unlikely to bias the results.

**4.2.4 English Channel.** In the English Channel, lumpfish were genetically differentiated from all other samples (Table 2). This pattern was also observed using microsatellite loci in a previous study by Whittaker et al. [45], which used the same individuals as in this study. As suggested therein, we agree that a plausible mechanism behind the southernmost population differentiation is warmer water (Table A in S1 Table), causing separation in spawning time and thus temporal isolation restricting gene-flow.

**4.2.5 Baltic Sea and the transition zone Skagerrak and Kattegat.** We confirm the previous observations [43, 45] of high genetic divergence of the Baltic Sea lumpfish and high similarity between individuals within the Baltic (Fig 4; Table A in S4 Table) and lower genome-wide variability (S3 Table). This is likely a combination of genetic drift due to limited number of founders and isolation, as well as strong diverging selection pressures in this marginal brackish environment, as has been shown for other species [17, 95, 96]. In fact, we show that variation in salinity together with temperature range, as well as minimum temperature with dissolved oxygen level, were significant factors explaining the genetic population structure of lumpfish. This was true even when the geographic variation was accounted for (Figure 5 in S1 File), indicative of local adaptation in Baltic lumpfish. Nonetheless, one individual of Baltic Sea genetic origin was found in the Skagerrak off-shore sample, despite the large difference in salinity between the two areas, suggesting that adult Baltic Sea lumpfish can survive in a much more saline environment than they inhabit.

The salinity transition zone between the North Sea and Baltic Sea showed surprisingly high levels of structuring. While the Skagerrak area, adjacent to the North Sea, was clearly clustered with the larger East Atlantic group (see below), the Kattegat formed its own genetic cluster. Individuals from Orust in the northernmost parts of Kattegat showed some degree of assignment to the Skagerrak cluster. However, all other samples in Kattegat, including samples next to the entrance of the Baltic Sea, were highly differentiated from both the Baltic Sea and Skagerrak (Fig 4). A similar pattern has previously been discovered in other widespread marine fish [17, 97, 98], implying the strong environmental gradient here could support very localized population structures.

**4.2.6 East Atlantic Group.** Several geographically distant samples from the Barents Sea, Northern and Southern Norway, Skagerrak, North Sea and the Outer Hebrides formed a large admixed East Atlantic group. The East Atlantic group showed weak but clear north-south genetic differences in the clustering analysis (S5 Table). Furthermore, several samples appeared to include individuals of different genetic backgrounds, suggesting existence of cryptic structures. This applied especially to the southwestern Norwegian fjords, where about half of the fish clustered with the East Atlantic group. Similar mixtures of fish with different genetic backgrounds were also detected in the Arctic including the Barents Sea, and to a lesser extent Svalbard, Namsen, and Alta. The majority of these samples were migrating lumpfish of unknown origin. Based on our results, we suggest that at least two lumpfish populations are mechanically mixed in the Arctic waters, and that the Arctic group could breed in the coastal areas close to the Barents Sea from where we did not have breeding fish available. Alternatively, the genetic division could have arisen from different breeding times or from assortative mating [99]. Similar cryptic genetic subdivisions have recently been described for northern shrimp (*Pandalus borealis* [100]) and polar cod (*Boreogadus saida* [101]. We ruled out the inherent weakness of clustering analyses favoring solution $K = 2$ [102, 103] as a plausible explanation for the observed two-cluster model in the Arctic, as well as for mainland Norway (see below) for several reasons: First, groupings were far from equal in size nor were they randomly distributed but constantly found within certain samples. Moreover, in the samples with several genetic groups, inbreeding coefficients were consistently higher than elsewhere (S3 Table), consistent with a Wahlund effect due to population subdivision.

It is still unclear if this large admixed East Atlantic group truly represents a single panmictic population–as previously suggested [45, 46, 104]–or if it is simply a group of mechanically mixed individuals outside their breeding season. Even though these samples are more connected than suggested by their spatial distance (S6 Table), our data supported some internal subdivision within the group (S5 Table; see results for hierarchical clustering for "Similar East Atlantic sites"). As STRUCTURE estimates both ancestral gene frequencies and admixture proportions for each individual in relation to other samples, interpretation of weakly differing admixture proportions can easily be misleading, and over-interpretation should be avoided [105].

**4.2.7 Norway.** Along the coast of Norway, we discovered two main groups: fish from large fjords in southwestern Norway; Sognefjord, Hardangerfjord and Boknafjord (Fig 4; S5 Table), and fish which clustered with the large East Atlantic group described above (Alta, Flatanger and Flekkefjord). The southwestern fjords had two highly differentiated genetic clusters, where all individuals were clearly assigned to one genetic group or another. We had detailed catch information of all fish from one of the fjords, Boknafjord (S2 File; Figure 7 in S1 File). Examination of the genetic clusters did not reveal any relationship with the status of the fish (juvenile vs adult) nor sex but when we compared the genetic groups with the geographic location of each fish, spatial division was evident. Within this large fjord, ~100 km in length and covering an area of 1579 km$^2$, fish that were caught closer to the open sea clustered with the large East Atlantic group, whereas the group found mainly deeper inside the fjord formed

another, distinct regional group (Figure 7 in S1 File). This is likely the same genetic group as Whittaker et al. [45] reported, who also found a distinct genetic group from this same region.

It is unclear what the regionally diverged coastal group in the southwestern fjords represents, but there are some possibilities: First, these fish could represent local lumpfish ecotypes. Fjord ecotypes have been described for other fish species such as Atlantic cod [106], and are likely upheld by innate differences in feeding and movement ecologies [107]. The ecotype hypothesis is supported by a recent field observation from Hardangerfjord, where lumpfish of all sizes were recorded during the breeding time when fish of intermediate sizes are not expected (C. Durif, personal observation). Secondly, it is also possible that this diverged group originates from another colonization event than the East Atlantic group. A third possibility is that this unique genetic group could be of farmed origin. Even being of wild origin, farmed lumpfish are mass-produced from a small number of broodfish. Founder effects, genetic drift and increased relatedness would skew population allele frequencies and could create distinct groupings [108, 109]. These large southwestern fjords are in the central salmon farming area in Norway, and escaped farmed lumpfish there is possible, and even likely [110]. However, a scenario where farmed fish of one distinct genotype would appear in such high numbers in several adjacent fjords simultaneously is unlikely. Broodstocks are primarily sourced from wild populations and thus unlikely to be this inbred.

## 4.3 Two datasets—Pitfalls and advantages

Targeting outlier loci that often reflect divergent selection processes can offer a powerful way to study species with large population sizes, high dispersal capabilities and frequent gene-flow leading to population homogenization [111–113] and has proven to be a highly effective approach in cases where random sets of markers show little or no structure [20, 114, 115]. Our two-step approach enabled a comparison of two types of datasets to resolve population structuring for lumpfish. The genome-wide dataset allowed us to detect outlier loci putatively indicative of local adaptation and simultaneously identify a smaller cost-effective set of loci to be genotyped in a large number of individuals with high geographic coverage [116]. The samples used in the initial 2b-RAD-analysis (see Table 1) and in the SNP selection phase were determined based on their location and previous knowledge of the genetic population structure for this species. This selection process has a few potential pitfalls: Studies conducted with a limited representation of the species' genome and based on few individuals can be prone to biases. The same applies to all later sampling stages if the samples used deviate from their source population (see e.g. [117–119]). Because genetic diversity is unequally distributed across populations, the populations in which SNPs are discovered may contribute to ascertainment bias. We tried to minimize known sources of biases in the marker development phase by including several sampling locations, using fish of both sexes and selecting markers with a wide genomic coverage. Whenever possible, samples from each site were collected at multiple time points and from a wider area. As RADseq is a non-selective process in relation to which genomic regions are to be included, the final genomic representation is most likely random. Thus, population processes mediated by neutral genetic markers, such as demographic history, can reliably be studied as they usually affect the entire genome. However, selection-related processes can be missed since selection often affects relatively small and targeted parts of the genome, many of which may not be included in random RADseqs [120]. Therefore, this study will tell only a partial and likely somewhat biased story of the lumpfish population structure, and more comprehensive studies–geographically and/or genome-wise–could provide additional insights.

However, outlier analysis remains a powerful approach to study larger patterns of differentiation. In this study, we found that the targeted dataset performed equally well or even better

than the genome-wide dataset at separating lumpfish populations regionally. This is despite the fact that only a small portion (3–17 SNPs; Table C in S2 Table) of the selected 139 SNPs were informative within each region. The targeted dataset displayed ~2 to 5 times higher level of divergence (Table 2), and greater clarity than the genome-wide dataset (Fig 3). In addition to the previously described population structure, our analyses revealed further genetic structure. Furthermore, our analyses revealed evidence of likely migrants and hybrids on both sides of the Atlantic.

## 4.4 Conclusions and management implications

Lumpfish occupy vast and highly variable marine environments and have high migration ability. However, based on its trophic level and long estimated population doubling time, lumpfish is expected to have low resilience to fishing pressure [121]. The impact of past and present fisheries on populations of lumpfish is largely unknown, and likely to be highly varying among regions and countries due to differences in management [25]. Moreover, the net productivity of lumpfish stocks may vary considerably between regions [122]. Significant declines in stock abundances have been reported in Canada, where lumpfish was designated as threatened in 2017 [123], whereas Norwegian and Barents Sea lumpfish stocks have lately been relatively stable or even increasing [124–126]. Our results show that lumpfish is a highly-structured species with local stocks which need to be managed as such although the stocks may mix physically outside of spawning season. Lumpfish is mainly targeted during the spawning season, which reduces the risk related to fisheries targeting mixed stocks [127]. Nevertheless, overfishing could still become a problem if not managed properly. Especially as earlier tagging studies have found lumpfish to display homing behavior [41, 42]. These studies were not, however, able to say if this homing behavior was to where the fish previously had spawned or where they themselves hatched, i.e., natal homing. The local adaptation and fine-scaled population genetic structure observed in this study suggest that there is little mixing of lumpfish between different areas across generations, supporting a hypothesis of natal homing. Consequently, if fishing pressure is too intense in a breeding population, it would not only remove the genetic material adapted to the local environment but repopulating overfished stocks may not be possible or at least take a very long time if natal homing is a prominent behavior in this species. More direct studies of homing across generations and experimental investigations of potential adaptive divergence are needed.

The use of lumpfish as a cleaner fish in commercial aquaculture has shown a rapid increase during the last decade, quickly becoming the most commonly used cleaner fish species for salmon farming [128]. Even though most lumpfish utilized in salmon farms are of farmed origin, broodstocks are still primarily sourced from wild fish. Currently, there are no regulations regarding the origin of cleaner fish released into the net-pens along the vast Norwegian coastline [128]. Although, there is evidence of lumpfish being shipped and translocated, the degree is not known [129]. Translocating fish can have many consequences for local populations, species and even ecosystems. Introduced organisms can bring foreign diseases and parasites novel to local ecosystems [130]. If the introduced organisms are genetically divergent from local populations, they may introduce unfavorable genetic material, which can result in altered population subdivision [30], reduced genetic variation, and/or reduced fitness [29, 131]. Local adaptation, together with the observed clear regional population subdivision likely upheld by spawning-site fidelity, emphasizes the need to manage the species on a regional level despite its wide transatlantic distribution. Considering on-going climate change for cold-adapted species like lumpfish, the standing genetic variation, especially in the southernmost latitudes of the species range might prove to be an invaluable resource in the future. The results from this study improve upon existing knowledge of lumpfish populations, which should be taken into

consideration for the future use and translocation of lumpfish to be used in the aquaculture industry. This study also provides a baseline for future monitoring of wild populations. The panel of SNPs used in this study has a high power to distinguish lumpfish from different populations and could in the future be applied for identifying wild and translocated fish.

## Supporting information

**S1 Checklist. Inclusivity in global research.**
(DOCX)

**S1 File. Figures.**
(DOCX)

**S2 File. Metadata.**
(XLSX)

**S3 File. Genome wide data set as vcf file.**
(VCF)

**S4 File. Targeted data set as txt file.**
(TXT)

**S5 File. PCA and DAPC figures.**
(DOCX)

**S6 File. LFMM2 outliers.**
(DOCX)

**S1 Table. Environmental parameters.**
(XLSX)

**S2 Table. SNP information.**
(XLSX)

**S3 Table. Basic population parameters.**
(XLSX)

**S4 Table. Pairwise $F_{ST}$ selected markers.**
(XLSX)

**S5 Table. Hierarchical clustering.**
(XLSX)

**S6 Table. Popgraph edges.**
(DOCX)

**S7 Table. Genomic outliers.**
(XLSX)

**S8 Table. Genetic outliers.**
(XLSX)

## Acknowledgments

We are grateful to those many people who helped us acquiring and handling the samples used in this study: Eva Farestveit, Elizabeth A. Fairchild, Vidar Wennevik, Gunnar Bakke, Lucilla Giulietti, John Fredrik Strøm, Jon-Ivar Westgaard, Ole Ingar Paulsen, Fredrik Staven, Håkan

Wennhage, Trond Rafoss, Lauri Pietikäinen, Johanne Gauthier, Kim Aarestrup, Kim Birnie-Gauvin, Carlos García de Leániz, Sofía Consuegra, Rasmus Hedeholm, Maia Olsen, Carina Jernberg, Hans Johannson, Curt Larsson, Kristian Nilsson, Rune Nilsen, Alain Fréchet, Sally Sherman, staff at the Centre for Sustainable Aquatic Research and the personnel on research vessels *Johan Hjort* and *G.O. Sars*. 2bRAD sequencing was performed by the SNP&SEQ Technology Platform in Uppsala. The facility is part of the National Genomics Infrastructure (NGI) Sweden and Science for Life Laboratory. The SNP&SEQ Platform is also supported by the Swedish Research Council and the Knut and Alice Wallenberg Foundation. The work was performed at the Institute of Marine Research, Bergen Norway (www.hi.no) and within the Linnaeus Centre for Marine Evolutionary Biology (www.cemeb.science.gu.se/). Earlier versions of this manuscript have previously been published in a thesis [132] and as a preprint in bioRxiv [133]. We are grateful for all the feedback and reviews we have received of this article and earlier versions thereof.

## Author Contributions

**Conceptualization:** Eeva Jansson, Ellika Faust, Carl André, Kevin A. Glover.

**Data curation:** Eeva Jansson, Ellika Faust, Dorte Bekkevold, Kim Tallaksen Halvorsen, Geir Dahle, Christophe Pampoulie, James Kennedy, Benjamin Whittaker, Laila Unneland, Søren Post, Carl André, Kevin A. Glover.

**Formal analysis:** Eeva Jansson, Ellika Faust, María Quintela.

**Funding acquisition:** Carl André, Kevin A. Glover.

**Investigation:** Eeva Jansson, Ellika Faust, Caroline Durif, Kim Tallaksen Halvorsen.

**Methodology:** Ellika Faust, Geir Dahle, Laila Unneland.

**Project administration:** Ellika Faust, Carl André, Kevin A. Glover.

**Resources:** Dorte Bekkevold, Carl André, Kevin A. Glover.

**Supervision:** Carl André, Kevin A. Glover.

**Validation:** María Quintela, Caroline Durif, Carl André.

**Visualization:** Eeva Jansson, Ellika Faust, María Quintela.

**Writing – original draft:** Eeva Jansson.

**Writing – review & editing:** Ellika Faust, Dorte Bekkevold, María Quintela, Caroline Durif, Kim Tallaksen Halvorsen, Geir Dahle, Christophe Pampoulie, James Kennedy, Benjamin Whittaker, Laila Unneland, Søren Post, Carl André, Kevin A. Glover.

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
