## [Decision Letter · Decision Letter 0]

27 Dec 2022

PONE-D-22-29672Global, regional, and cryptic population structure in a high gene-flow transatlantic fishPLOS ONE

Dear Dr. Faust,

Thank you for submitting your manuscript to PLOS ONE. As you will see from the reviewers' suggestions, some minor modifications are required. Therefore, we invite you to submit a revised version of the manuscript that addresses the points raised during the review process.

We look forward to receiving your revised manuscript.

Kind regards,

Roberta Cimmaruta, PhD

Academic Editor

PLOS ONE

Journal Requirements:

5. We note that Figures 1 and 4 in your submission contain map images which may be copyrighted. All PLOS content is published under the Creative Commons Attribution License (CC BY 4.0), which means that the manuscript, images, and Supporting Information files will be freely available online, and any third party is permitted to access, download, copy, distribute, and use these materials in any way, even commercially, with proper attribution. For these reasons, we cannot publish previously copyrighted maps or satellite images created using proprietary data, such as Google software (Google Maps, Street View, and Earth). For more information, see our copyright guidelines: http://journals.plos.org/plosone/s/licenses-and-copyright.

 a. You may seek permission from the original copyright holder of Figures 1 and 4 to publish the content specifically under the CC BY 4.0 license. 

Reviewers' comments:

Reviewer's Responses to Questions

**Comments to the Author**

1. Is the manuscript technically sound, and do the data support the conclusions?

Reviewer #1: Yes

Reviewer #2: Partly

2. Has the statistical analysis been performed appropriately and rigorously? 

Reviewer #1: Yes

Reviewer #2: Yes

3. Have the authors made all data underlying the findings in their manuscript fully available?

Reviewer #1: Yes

Reviewer #2: No

4. Is the manuscript presented in an intelligible fashion and written in standard English?

Reviewer #1: Yes

Reviewer #2: Yes

5. Review Comments to the Author

Reviewer #1: General comments,

The manuscript entitled 'Global, regional, and cryptic population structure in a high gene-flow transatlantic fish' aimed to define the population structure at small-scale in a commercially exploited fish widely distributed based on genome wide SNP markers in a representative set of individuals (95) and targeted SNPs putatively subjected to selection in a significative sample size covering the whole species distribution (1,669).

The manuscript would be of interest for a wide audience related to evolutionary genetics but also for those involved in fishery management and conservation.

The manuscript is well written, data very well analyzed and results are sound. Even tough I enjoyed pretty much the reading of the manuscript, the discussion is exhaustive and hard to follow, it lacks also of a detailed analysis of the results from adaptive data in relation with potential variables acting on generating local adaptations and genetic divergence. The discussion seems to me a re-count of the results in most of the text, with some exceptions, I would recommend to discuss deeply those aspects in order to highlight the role of environmental factors in a wide variability of habitats and the species adaptive potential, in summary, a more evolutionary background would be very welcomed as this is the main contribution of the manuscript and would facilitate the understanding in a more comprehensive form the findings.

I consider the manuscript acceptable for publication in PLOS after dealing with the suggestions made if authors consider them valuable.

Minor comments

Line 100. Please specify if it was documented based on tagging studies

Line 103. Be more specific about the genetic markers used (e.g. type and number of loci)

Line 319. Table 2 legend, In methods says that were used 2000 bootstraps, please clarify.

Line 322. Please provide details on the criteria used to define groups for the AMOVA.

Line 369. Figure 3 is not easy to interpret and would seems that legend is not describing accurately what is in the figure, is not clear the K and the figure for each dataset.

Figure 5. It would be possible to overlap this graph on the locations map?

Lines 464-465. Natal homing may be linked to adaptive process, please give a brief discussion about this.

Lines 487-489

Lines 495-497. I don´t see enough support for this conclusion without tagging data, the term natal homing can be applied once is determined the repeated use of the sites across generations and can be confounded with site fidelity (not related to reproduction) in abscence of tagging data.

Lines 519-520. If Isfjorden is a relict population I would suggest to assess past reductions in population size for this population and discuss deeper this possibility as there could be ancestral polymorphism causing those similarities or a strong adaptive factor.

Lines 675-677. Please specify which would be the difference in considering management units based on both, genome wide data and targeted dataset as both may have different implications on preserving gene diversity and population viability. Also discuss the conservation implications of detecting a more detailed regional sub-structuring.

Reviewer #2: This work is a valuable effort for improving the knowledge on the highly impacted lumpfish. The amount of work to produce and analyse the dataset is evident, but some issues need to be addressed in order to improve the clarity and the coherence of the manuscript and to have robust inference from the molecular data produced. Please find below a few main ones listed, while all comments are provided in the PDF attached.

From Table 1 is evident the huge sampling effort, but also the highly variable range of individuals collected, both in terms of sampling years (samples are scattered across approx. 10 years) as well as multiple life stages and unbalanced sex ratio, and other life history traits (migratory, spawning). I am fully aware that sampling marine fish is not a task that can be pre-set, but I would appreciate the authors being aware and discuss the consequences of this sampling design of the results.

Did the authors attempted to assess differentiation pattern using only homogeneous samples, i.e., same life stage, in particular only spawning adults? I think this is quite relevant.

Authors state several times that “targeted dataset performed equally well or even better than the genome-wide dataset at separating lumpfish populations regionally”. I think this is not something to be claimed, since the targeted dataset was cherrypicked with the purpose of maximizing the differentiation. I see this point as a circular reasoning and a weakness of the work.

The same authors worked on Aquaculture‐ mediated translocation and hybridization in lumpfish. How their previous works relate to this one? And how the described translocation and hybridization is accounted for in discussing the differentiation pattern obtained in this study? For example, in explaining the large admixed East Atlantic group.

I think there is no associated code with this manuscript. Raw reads have been submitted to a BioProject, but no other info or data are available. This is a really sound and comprehensive bioinformatics paper and I recommend publication of all manuscript related scripts and datasets (made available via GitHub or similar).

Keep colors consistent across figures. In several figures samples are color-coded based on their geographic region, but colors are different among figures.

Please revise the Supplementary Materials since the authors refer often to differential formatting to highlight data (bold, yellow background, gray background). Unfortunately, this formatting is not always precise, accurate and coherent with the text and captions. I.e., Supplementary Table 8: “SNPs (2) in gray background were also picked as significant in genome-wide analysis of outliers” but 3 loci are shown with gray background. Several other similar occurrences are presents in the Supplementary Materials.

6. PLOS authors have the option to publish the peer review history of their article (what does this mean?). If published, this will include your full peer review and any attached files.

Reviewer #1: No

Reviewer #2: No

---

## [Author Response · Author response to Decision Letter 0]

7 Feb 2023

>> We are very grateful to the Editor and the two Reviewers for their encouraging comments and suggestions. We have replied to the comments on a point-by-point basis below. Each answer starts with ">>".

Reviewer #1: General comments,

The manuscript entitled 'Global, regional, and cryptic population structure in a high gene-flow transatlantic fish' aimed to define the population structure at small-scale in a commercially exploited fish widely distributed based on genome wide SNP markers in a representative set of individuals (95) and targeted SNPs putatively subjected to selection in a significative sample size covering the whole species distribution (1,669).

The manuscript would be of interest for a wide audience related to evolutionary genetics but also for those involved in fishery management and conservation.

The manuscript is well written, data very well analyzed and results are sound. Even tough I enjoyed pretty much the reading of the manuscript, the discussion is exhaustive and hard to follow, it lacks also of a detailed analysis of the results from adaptive data in relation with potential variables acting on generating local adaptations and genetic divergence. The discussion seems to me a re-count of the results in most of the text, with some exceptions, I would recommend to discuss deeply those aspects in order to highlight the role of environmental factors in a wide variability of habitats and the species adaptive potential, in summary, a more evolutionary background would be very welcomed as this is the main contribution of the manuscript and would facilitate the understanding in a more comprehensive form the findings.

>> We are grateful to reviewer 1 for their thorough and careful consideration of the manuscript. Although we found strong associations with environmental variables, suggesting local adaptation, our genetic data only covers a limited part of the genome and thus only shows a snapshot of the variation in this species. Furthermore, in order to claim that the patterns we see are a result of adaptive divergence, experimental validation is needed (e.g., in the lab or translocations experiments), or at least data on phenotypic divergence which is associated with local adaptation. Because of these reasons, we did not want to speculate too deeply on this topic. We did however add a section on this topic in section 5.4 “Conclusion and management implications”. Moreover, many aspects of lumpfish basic ecology are still unknown. We don't know for instance if there are different migratory tendencies. We discuss this in our manuscript e.g., in connection with cryptic population structure. We believe it to be relevant to discuss the patterns we observe for each region separately, which does require a short summary of the results for context. We have tried to compensate for this by expanding the discussion in the directions suggested by the two reviewers.

I consider the manuscript acceptable for publication in PLOS after dealing with the suggestions made if authors consider them valuable.

Minor comments

Line 100. Please specify if it was documented based on tagging studies

>> Yes, this is based on tagging studies. We have now specified that in the text. “However, tagging studies have also shown that facultative iteroparity is possible, and that females which return to spawn the following year, spawn at roughly the same time [41] and in the same area [42] as they did previously, in support of homing behavior.”

Line 103. Be more specific about the genetic markers used (e.g. type and number of loci)

>> We have adjusted the text accordingly “Earlier studies, using 10-11 microsatellite loci, ... ”

Line 319. Table 2 legend, In methods says that were used 2000 bootstraps, please clarify.

>> It should be 2000 bootstraps. We have now corrected the typing error accordingly.

Line 322. Please provide details on the criteria used to define groups for the AMOVA.

>> Samples are all individuals collected at the same site at the same time point and the regions reflect the geographic proximity of sampling sites, as described in table 1. We have now clarified the legend of the AMOVA table.

Line 369. Figure 3 is not easy to interpret and would seems that legend is not describing accurately what is in the figure, is not clear the K and the figure for each dataset.

>> We have now clarified in the figure legend for figure 3 that the top panel refers to the genome-wide dataset with K=7 and the bottom panel refers to the targeted dataset with K=9.

Figure 5. It would be possible to overlap this graph on the locations map?

>> Yes, we have now added a map background.

Lines 464-465. Natal homing may be linked to adaptive process, please give a brief discussion about this.

>> We have added some further discussion on homing and adaptive processes to the discussion, especially in sections 5.2 and 5.4

Lines 495-497. I don´t see enough support for this conclusion without tagging data, the term natal homing can be applied once is determined the repeated use of the sites across generations and can be confounded with site fidelity (not related to reproduction) in abscence of tagging data.

>> Earlier tagging studies have noted some homing behavior / site fidelity. However, they were not able to say if this homing behaviour was only to where the fish had spawned previously or where they themselves hatched (natal homing). In places where we have inaccurately referred to these tagging studies as in reference for “natal” homing, we have now changed the wording to “homing behavior” or “site fidelity”. However, we do still argue that the local population genetic structure we observe would suggest that there is little mixing of lumpfish between different areas across generations, which could support the hypothesis of natal homing. Although more direct studies investigating homing across generations is needed to fully confirm this hypothesis. 

Lines 519-520. If Isfjorden is a relict population I would suggest to assess past reductions in population size for this population and discuss deeper this possibility as there could be ancestral polymorphism causing those similarities or a strong adaptive factor.

>> We agree with the reviewer that this is a very likely scenario. Unfortunately, we only have data from the targeted dataset of Isfjorden which makes it difficult, and likely also biased, to analyze the demographic history, such as past changes in population size, of this population. But we do believe that this population presents an interesting avenue for future research on the evolutionary history of this species. We have also added these reflections to this paragraph. 

Lines 675-677. Please specify which would be the difference in considering management units based on both, genome wide data and targeted dataset as both may have different implications on preserving gene diversity and population viability. Also discuss the conservation implications of detecting a more detailed regional sub-structuring.

>> We don’t find that the two datasets are in any way contradictory. The targeted dataset simply fills geographic gaps in the genome-wide dataset. If anything, this suggests that management should try and avoid extrapolation of population/stock data from one geographic region to another, as there are many local differences. We have added a short section on the conservation implications of regional sub-structuring under “Conclusions and management implications”

Reviewer #2: 

This work is a valuable effort for improving the knowledge on the highly impacted lumpfish. The amount of work to produce and analyse the dataset is evident, but some issues need to be addressed in order to improve the clarity and the coherence of the manuscript and to have robust inference from the molecular data produced. Please find below a few main ones listed, while all comments are provided in the PDF attached.

From Table 1 is evident the huge sampling effort, but also the highly variable range of individuals collected, both in terms of sampling years (samples are scattered across approx. 10 years) as well as multiple life stages and unbalanced sex ratio, and other life history traits (migratory, spawning). I am fully aware that sampling marine fish is not a task that can be pre-set, but I would appreciate the authors being aware and discuss the consequences of this sampling design of the results.

Did the authors attempted to assess differentiation pattern using only homogeneous samples, i.e., same life stage, in particular only spawning adults? I think this is quite relevant.

>> We thank reviewer 2 for the thorough and careful consideration of the manuscript. They are correct in that the sampling scheme was opportunistic and as such we have an unequal representation of geographic regions, sample sizes, age groups, life stages, sex, spawning or migratory fish, and time of sampling. We performed several analyses where we separated juveniles from adults, males from females and migratory from spawning fish, but we found very little in terms of links to the observed patterns of population structure. We did however see that samples with only breeding and juvenile lumpfish formed regional more clear-cut groups, whereas migrating fish, mid-sea samples and samples known to contain both juveniles and adults were more admixed. We do describe these analyses and potential biases in several places in the manuscript, e.g. concerning age and life history stage (chapter 5.2), sex bias (chapter 5.2.3), cryptic clusters in Southwestern Norwegian fjords in terms of life history and sex (chapter 5.2.7) and chapter 5.3 which handles several aspects of potential bias and how we tried to deal with them.

Authors state several times that “targeted dataset performed equally well or even better than the genome-wide dataset at separating lumpfish populations regionally”. I think this is not something to be claimed, since the targeted dataset was cherrypicked with the purpose of maximizing the differentiation. I see this point as a circular reasoning and a weakness of the work.

>> We only found one such mention (section 5.3), but we understand the reviewer's concern, as it is very much expected to find a higher divergence between regions when only looking at outlier loci for the specified regions. The reason for us stating something which at first glance might seem obvious is that 1) only a very small number of SNPs (3-17 of the 139 SNPs) were informative within each region and 2) the targeted dataset could separate several regions which were not part of the genome-wide data set or SNP selection. It was in this context we wanted to highlight the targeted data set's ability to separate populations.

The same authors worked on Aquaculture‐ mediated translocation and hybridization in lumpfish. How their previous works relate to this one? And how the described translocation and hybridization is accounted for in discussing the differentiation pattern obtained in this study? For example, in explaining the large admixed East Atlantic group.

>> We presume the reviewer is referring to some authors' previous work on translocation and hybridisation in wrasse. Although both species are used as cleaner fish, the similarities stop there in terms of their biology, distribution and how they are sourced. Of course, there is a possibility that some of the admixture does stem from lumpfish escaping and mixing. We discuss the possibility that escapees could explain the cryptic clusters in the Southwestern Norwegian fjords (chapter 5.2.7). However, this is hard to disentangle without any proper baseline from before the use of lumpfish as cleaner fish. A hope is that this study will provide such a baseline as well as a SNP panel which can be used for future identification purposes. 

I think there is no associated code with this manuscript. Raw reads have been submitted to a BioProject, but no other info or data are available. This is a really sound and comprehensive bioinformatics paper and I recommend publication of all manuscript related scripts and datasets (made available via GitHub or similar).

>> We have now added SNP data and metadata as supplementary material, as well as added links to relevant GitHub pages where scripts and the pipeline can be found.

Keep colors consistent across figures. In several figures samples are color-coded based on their geographic region, but colors are different among figures.

We have changed the colours in Figures 1 and 6 to match the colours in figure 5, which roughly correspond to structure colours in figure 3. We have also added the same colours to figure 2 for easier interpretation.

Please revise the Supplementary Materials since the authors refer often to differential formatting to highlight data (bold, yellow background, gray background). Unfortunately, this formatting is not always precise, accurate and coherent with the text and captions. I.e., Supplementary Table 8: “SNPs (2) in gray background were also picked as significant in genome-wide analysis of outliers” but 3 loci are shown with gray background. Several other similar occurrences are presents in the Supplementary Materials.

The legend of Supplementary Table 8 is accurate, although 3 cells are highlighted in grey, 2 of them are the same loci, as this locus was picked up by two independent variables. (“Bolded SNPs were significantly related to variation of several independent environmental variables.”). We have now gone over all supplementary material to ensure that the texts accurately describe the tables and the figures.

Responses to comments in PDF:

Line 45: not only on this data, many others are included in stock assessments

>> We have now added the other two main data forms catch and demography changes to the sentence

Line 49: check references bracket formatting

>> We have now checked the reference formatting and corrected int where inaccurate

Line 87: bracket missing?

>> Fixed

Line 138: Bold codes are not visible

>> Thank you for pointing this out, we have now corrected this

Line 150: I think there is no associated code with this manuscript. Raw reads have been submitted to a BioProject, but no other info or data are available. This is a really sound and comprehensive bioinformatics paper and I recommend publication of all manuscript related scripts and datasets (made available via GitHub or similar)

>> We have now added SNP data and metadata as supplementary material, as well added links to relevant GitHub pages where scripts and the pipline can be found.

Line 153: please provide more details about technical replicates (number of library employed, geographical range, etc etc).

>> We have now added this information to the text.

Line 163: I understand this is an arbitrary threshold that has to be set, but please argument why this value was selected

>> The aim was to have 20-30 SNPs from each pairwise comparison. But when designing primers to be pooled in assays, many SNPs may have to be thrown out as their primers are too similar or clash in some ways. so here the target was to have at least twice as many SNPs to choose from. Using this cut of 0.4 we had roughly 70-100 SNPs from each pairwise comparison.

Line 172: TableS2.b since S.Table 2c. List of regionally most divergent SNPs. Loci were the ones with PCA loading above the threshold of 0.1 within each given region. 

>> We have changed this accordingly

Line 304: please explain this statement. The data presented here are the 4393 SNPs.

>> This is a mistake and should have been part of the later section referring to the higher FST in the targeted dataset.

Line 305: which ones? not clear from Supplementary Table 3, where all but one HWE = overall p-value for Hardy-Weinberg equilibrium test are zero

>> Significantly deviating heterozygosities are those where confidence intervals for Fis do not include negative values but are positive in their whole 95% CI range. HWE p-values were measured over all 139 SNPs and were extremely small in all populations because some loci were highly deviating in each case. This is likely because loci that differentiate regions were selected and some of them are possibly also under selection. Here some extreme values skew the result so much that we decided to remove this column for HWE p-value from the table as it doesn't bring any additional value and causes confusion. 

Line 308: please include the same coding used in Table1 to facilitate the cross reference (Location Code Region)

>> We have now added the coding for the sites to table 2

Line 331: check

>> we have now removed the “of” from this sentence

Line 356: two or three?

>> It should be three and has now been corrected.

Line 374: please assess if the samples which were genotyped with both methods show a coherent clustering in the two STRUCTURE analysis

>> We do see that the genotypes show concurrent clustering between the two methods when comparing the STRUCTURE analysis. Samples which were genotyped with both methods are indicated by the black lines in Figure 3. The main difference is that when using the targeted SNPs the distinction between clusters became clearer. This is also visible in S5 File on pages 2-3 where we display PCA clustering of only the individuals genotyped with both methods, with some difference in explanatory percentage, as would be expected when using 139 SNPs instead of ~4k. Similarly, we also show pairwise FST values only using these individuals in Table B in the S4 Table. Although the FST estimates have increased overall when using the targeted SNPs (as would be expected), the increase is overall even for all pairwise comparisons. Meaning that pairwise estimates that are low with the genome-wide data are also low when using the 139SNP, and estimates that are high with the genome-wide data are also high when using the 139SNP. Using the same individuals or more does not alter this pattern to any larger degree.

Line 422: loci

>> Corrected in the text

Line 442: Is the locus annotated?

>> Yes, all annotations are given in Table A in S2 Table. This locus was not inside any gene but close to some. You can see these under column "features".

Line 492-495: did the authors attempted to assess differentiations pattern using only homogeneous samples (same life stage)? I think this is quite relevant 

>> In the cases when we did have information on life stages, we did attempt to asses differentiation among these. Unfortunately, for many samples, we did not have this information. See also the response to reviewer 1.

Line 600: Supplementary Table 5 is mentioned and referred to so many times that probably is worth to include it in the main text.

>> Supplementary Table 5 is a very large file containing the method, result and evaluation of hierarchal clustering. Figure 4 displays a summary of this rather complex table and they are often referenced together when relevant. We added a reference to this figure in the text in addition to referencing the Supplementary Table.

Line 635: not clear how many outlier in the genome-wide dataset are also included in the targeted dataset

>> Outliers in the genome-wide dataset which are also in the targeted dataset are visible in table B and C in the S7 Table under the column “name_if_in_genotyping_panel”. Loci which were part of the panel but were filtered out due to lack of amplification are denoted as “(no amp.)”

Line 658: I think this is not something to be claimed, since the targeted dataset was cherrypicked with the purpose of maximizing the differentiation. I see this point as a circular reasoning and a weakness of the work.

>> We understand Reviewer 2s concern, as it is very much expected to find a higher divergence between regions when only looking at outlier loci for the specified regions. The reason for us stating something which at first glance might seem obvious is that 1) only a very small number of SNPs (3-17 of the 139 SNPs) were informative within each region and more importantly 2) the targeted dataset could separate several regions which were not part of the genome-wide data set or SNP selection. It was in this context (see the following sentence in the manuscript) we wanted to highlight the targeted data set's ability to separate populations.

---

## [Editor Report · Decision Letter 1]

7 Mar 2023

Global, regional, and cryptic population structure in a high gene-flow transatlantic fish

PONE-D-22-29672R1

Dear Dr. Faust,

We’re pleased to inform you that your manuscript has been judged scientifically suitable for publication and will be formally accepted for publication once it meets all outstanding technical requirements.

Kind regards,

Roberta Cimmaruta, PhD

Academic Editor

PLOS ONE
---

## [Editor Report · Acceptance letter]

12 Mar 2023

PONE-D-22-29672R1 

Global, regional, and cryptic population structure in a high gene-flow transatlantic fish 

Dear Dr. Faust:

I'm pleased to inform you that your manuscript has been deemed suitable for publication in PLOS ONE. Congratulations! Your manuscript is now with our production department. 

Kind regards, 

on behalf of

Professor Roberta Cimmaruta 

Academic Editor

PLOS ONE